# Distributionally Robust Classification for Multi-source Unsupervised Domain Adaptation

**Seonghwi Kim**
Pohang University of Science and Technology
kshwi@postech.ac.kr

**Sung Ho Jo**
Pohang University of Science and Technology
tjdgh1813@postech.ac.kr

**Wooseok Ha**
Korea Advanced Institute of Science and Technology
haywse@kaist.ac.kr

**Minwoo Chae**[*]
Pohang University of Science and Technology
mchae@postech.ac.kr

## Abstract

Unsupervised domain adaptation (UDA) is a statistical learning problem when the distribution of training (source) data is different from that of test (target) data. In this setting, one has access to labeled data only from the source domain and unlabeled data from the target domain. The central objective is to leverage the source data and the unlabeled target data to build models that generalize to the target domain. Despite its potential, existing UDA approaches often struggle in practice, particularly in scenarios where the target domain offers only limited unlabeled data or spurious correlations dominate the source domain. To address these challenges, we propose a novel distributionally robust learning framework that models uncertainty in both the covariate distribution and the conditional label distribution. Our approach is motivated by the multi-source domain adaptation setting but is also directly applicable to the single-source scenario, making it versatile in practice. We develop an efficient learning algorithm that can be seamlessly integrated with existing UDA methods. Extensive experiments under various distribution shift scenarios show that our method consistently outperforms strong baselines, especially when target data are extremely scarce.

## 1 Introduction

Many supervised learning algorithms rely on the assumption that the training and test data are drawn from the same underlying distribution. However, this assumption is often violated in real-world applications due to various factors such as environmental changes, sampling biases, or temporal dynamics, leading to distribution shifts between training and deployment environments (Beery et al., 2018; Zech et al., 2018; Volk et al., 2019). Such mismatches can result in significant performance degradation, posing a fundamental challenge to achieving reliable generalization in practice (Gulrajani & Lopez-Paz, 2020; Koh et al., 2021; Sagawa et al., 2021).

Among various challenges posed by distribution shifts, we focus on unsupervised domain adaptation (UDA; Ganin et al. (2016); Long et al. (2015; 2018); Ben-David et al. (2010)). In UDA, we are given labeled data from a source domain and unlabeled data from a target domain, where the distributions of the two domains are related but not necessarily identical. The discrepancy may arise from differences in the marginal distribution of the input (Shimodaira, 2000; Sugiyama et al., 2007), the conditional distribution of the output given the input (Jin et al., 2023; Cai et al., 2023), or both (Tachet des Combes et al., 2020; Wu et al., 2025). The central objective is to build a predictive model that generalizes to the target domain by effectively transferring knowledge from the source domain.

Two representative approaches in UDA are distribution alignment and pseudo-labeling methods. Distribution alignment reduces the discrepancy between source and target domains, either in the

---

[*]Corresponding author.

data space (Courty et al., 2017b) or in the feature space (Ganin et al., 2016; Long et al., 2018). In contrast, pseudo-labeling methods refine target predictions using a source-trained model (Amini & Gallinari, 2003; Kumar et al., 2020; Wei et al., 2020), inspired by early work on self-training (Lee, 2013). Despite their popularity, both distribution alignment and pseudo-labeling methods often face practical limitations. Two common scenarios are when the amount of unlabeled target data is limited, or when the source data contains spurious correlations. When target data are scarce, alignment estimates can become unreliable and pseudo-labels may be noisy, which can lead to poor generalization (Liang et al., 2020; Xie et al., 2020). When spurious correlations are present—for example, when irrelevant features like background or color are strongly associated with labels—models often rely on these shortcuts, which do not transfer to the target domain and ultimately degrade performance (Arjovsky et al., 2019; Zhao et al., 2019; Liu et al., 2021).

In this paper, we propose a new method for UDA, with a particular focus on classification tasks, based on the framework of distributionally robust learning (also known as distributionally robust optimization; DRO[1]). Our approach is inspired by the maximin effect estimation method developed for regression problems with multi-source data (Meinshausen & Bühlmann, 2015). While originally motivated by the multi-source setting, our method can also be applied in single-source scenarios by simulating multiple pseudo-sources through random indexing.

The proposed method estimates the conditional distribution of the output given the input for each source domain and constructs an ambiguity set consisting of their mixtures. The mixing weights are allowed to vary, enabling the model to adjust how much it trusts each source, while the input distribution is permitted to shift within a small Wasserstein ball around the observed target inputs. By explicitly modeling both sources of uncertainty—(i) which conditionals to rely on and (ii) how the target inputs may vary—our method improves target prediction performance, particularly when target data are scarce or when source domains contain spurious correlations.

We empirically evaluate our approach on widely used benchmarks, including digit classification tasks (MNIST, SVHN, USPS) and spurious correlation datasets (Waterbirds, CelebA, Colored MNIST). Our experiments focus on two key scenarios: (1) standard UDA with limited unlabeled target data and (2) settings where spurious correlations hinder generalization. In both cases, our method substantially outperforms existing approaches in domain adaptation and robust learning.

Our main contributions can be summarized as follows:

- We propose a novel distributionally robust framework that jointly models uncertainty in the target covariate distribution and the conditional label distribution defined in the feature space.
- We develop a tractable minimax optimization algorithm for the proposed method, which can be seamlessly combined with existing UDA methods.
- We provide extensive experiments demonstrating substantial improvements over well-known baselines in two challenging scenarios: target data scarcity and spurious correlations.

In the following, we review key lines of related work, including modern approaches to UDA, DRO-based methods, and other robust methods addressing spurious correlations.

## 1.1 RELATED WORKS

**Unsupervised domain adaptation** UDA has been applied in a wide range of areas, including computer vision (Hoffman et al., 2018), natural language processing (Gururangan et al., 2020), and healthcare (Kamnitsas et al., 2017). Popular approaches in UDA learn domain-invariant features by aligning marginal or class-conditional distributions (Long et al., 2015; Ganin et al., 2016; Sun & Saenko, 2016; Tzeng et al., 2017; Long et al., 2018), or by separating normalization statistics across domains (Chang et al., 2019). They also regularize decision boundaries (Shu et al., 2018) and exploit pseudo-labels—including source-free variants (Liang et al., 2020; Yang et al., 2021). Optimal-transport methods align either marginal input distributions (Courty et al., 2017b) or joint distributions (Courty et al., 2017a; Damodaran et al., 2018). Recent work explores concept-based interpretability, diffusion-based alignment, information-theoretic regularization, one-shot settings, and heterogeneous-modal adaptation (Peng et al., 2024; Xu et al., 2025; Tan et al., 2025; Zhang

---

[1]While DRO was originally studied in the context of optimization, in this paper we use the term more broadly to refer to distributionally robust learning methods under distribution shift.

et al., 2025; Yang et al., 2025). Causal-representation-based approaches (Kong et al., 2023; Li et al., 2023) address multi-source settings by identifying invariant latent structures across heterogeneous environments, offering a conceptually different direction within UDA. However, these methods can fail in the presence of spurious correlations.

To address this limitation, recent studies attempt to disentangle causal and spurious features using unlabeled target data. Examples include counterfactual consistency, generative interventions, and self-training refinements (Chen et al., 2020; Yue et al., 2023; Wei et al., 2025). Similar approaches have also been explored in multi-source settings (Li et al., 2023; Liu et al., 2025). However, these methods often rely on strong structural assumptions or are highly sensitive to the quality of initial pseudo-labels, making them fragile under scarce or imbalanced target data.

**Distributionally robust optimization** Distributionally robust framework has recently gained attention in various areas, including UDA. One line of work in DRO defines ambiguity sets via Wasserstein balls or $f$-divergences, which provide robustness to small perturbations of the source distribution (Ben-Tal et al., 2013; Duchi et al., 2021; Awad & Atia, 2023). Another line considers maximin-effect or GroupDRO formulations that address mixture shifts, where the target distribution differs in mixing proportions across subpopulations (Meinshausen & Bühlmann, 2015; Zhan et al., 2024; Wang et al., 2025; Guo, 2024; Sagawa et al., 2019). Maximin-effect approaches are closely related to our setting but have been mostly studied in regression tasks. We provide further discussion on this point in Section 3. Mixture-shift formulations, particularly GroupDRO, are also relevant when spurious correlations arise from changing group proportions. However, GroupDRO typically assumes access to group labels and does not explicitly use unlabeled target data for improving target prediction performance. (Guo et al., 2025) studies DRO for classification, but this paper mainly focuses on statistical convergence rates and the construction of bias-corrected estimators. More recently, Jo et al. (2026) introduced a distributionally robust learning framework using hierarchical ambiguity sets to specifically address subgroup shifts.

**Other robust methods under spurious correlation** A closely related setting to UDA is domain generalization (DG), where unlike UDA, no unlabeled target data are available during training. Representative DG methods, such as IRM (Arjovsky et al., 2019), V-REx (Krueger et al., 2021), and Fish (Shi et al., 2021), aim to learn representations that are invariant across multiple training environments to mitigate spurious correlations. Beyond domain generalization, other lines of work address spurious correlations more directly. These include approaches such as sample reweighting, biased auxiliary models, and adversarial weighting without group labels (Nam et al., 2020; Lahoti et al., 2020; Sohoni et al., 2020; Liu et al., 2021). Additional strategies suppress spurious features through contrastive objectives or mixup-based regularization (Zhang et al., 2022; Jin et al., 2024).

## 2 PRELIMINARIES

This section introduces basic notations, definitions and some preliminary setups. Let $(X, Y)$ be the random variables of the input and output pair, where $X \in \mathcal{X} \subseteq \mathbb{R}^d$ and $Y \in \mathcal{Y}$. For a joint distribution $P$ of $(X, Y)$, its marginal distribution of $X$ and conditional distribution of $Y$ given $X$ are denoted by $P_X$ and $P_{Y|X}$, respectively. Also, we use the lower case $p$ for denoting the density of the corresponding distribution denoted by the upper case $P$, and vice versa. The expectation with respect to the distribution $P$ is denoted by $\mathbb{E}_P$.

Suppose for a moment that there is a single source domain. Let the population distributions of the source and target domains be denoted by $P^{\mathrm{sc}}$ and $P^{\mathrm{tg}}$, respectively. For a function $f^\theta$ parameterized by $\theta \in \Theta \subseteq \mathbb{R}^p$, let $\ell(f^\theta(X), Y)$ be the loss function, which is the cross-entropy loss in most cases. In UDA, the goal is to minimize the target risk $\mathbb{E}_{P^{\mathrm{tg}}}[\ell(f^\theta(X), Y)]$ with the labeled source and unlabeled target data. The empirical risk minimizer (ERM) trained on the source data is the most naive baseline in UDA. However, when $P^{\mathrm{sc}}$ and $P^{\mathrm{tg}}$ are different, ERM can exhibit poor predictive performance on the target domain.

An important alternative to ERM estimator under distributional shift is DRO. Let $\mathcal{Q}$ denote a class of joint distributions over $(X, Y)$, often referred to as the ambiguity set. In general, the DRO estimator with ambiguity set $\mathcal{Q}$ can be obtained by solving the minimax optimization problem

$$\underset{\theta \in \Theta}{\text{minimize}} \ \sup_{Q \in \mathcal{Q}} \mathbb{E}_Q[\ell(f^\theta(X), Y)]. \tag{1}$$

Constructing a suitable ambiguity set $\mathcal{Q}$ is a crucial and central component of DRO methods. A common approach is to define $\mathcal{Q}$ as a small (pseudo-)metric ball around the empirical distribution of source data or one of its variants. In practice, a key consideration is that the inner supremum must be not only well-defined but also computationally tractable, as it directly affects the overall optimization. For this reason, popular choices for the underlying metric include the Wasserstein distance (Gao et al., 2024; Gao & Kleywegt, 2023; Mohajerin Esfahani & Kuhn, 2018; Blanchet et al., 2021) and $f$-divergences (Duchi & Namkoong, 2021; Namkoong & Duchi, 2016), which allow for efficient approximations or closed-form solutions in many settings.

While ambiguity sets around the source distribution provide robustness to minor shifts, they are less suitable for structured shifts. A notable case is subpopulation shift, where the source is a mixture of subpopulations and the target has the same subpopulations but different mixing weights. Here, the target can be far from the source in standard distances, making small-ball sets inadequate. A crucial example of an alternative formulation is the ambiguity set used in GroupDRO (Sagawa et al., 2019), which is specifically designed to address subpopulation shifts.

## 3 PROPOSED DISTRIBUTIONALLY ROBUST LEARNING METHOD

In this section, we present our multi-source DRO framework together with the corresponding learning algorithm. We assume access to labeled multi-source datasets $\mathbf{D}^{(k)} = \{(x_i^{(k)}, y_i^{(k)})\}_{i=1}^{N^{(k)}}$, where $\mathbf{D}^{(k)}$ denotes the dataset from the $k$th source domain ($k = 1, \ldots, K$), and an unlabeled target dataset $\mathbf{D}^{\text{tg}} = \{x_i^{\text{tg}}\}_{i=1}^{N^{\text{tg}}}$. Although the framework is motivated by the multi-source setting, it can also be applied to single-source problems, as discussed in Section 3.1. Section 3.2 provides a high-level formulation of the proposed ambiguity set, highlighting the core intuition behind our method. Section 3.3 specifies the components of the ambiguity set in greater detail, and Section 3.4 presents the detailed learning algorithm.

### 3.1 MULTI-SOURCE FRAMEWORK FOR SINGLE-SOURCE PROBLEMS

Suppose that the source dataset $\mathbf{D}^{\text{sc}} = \{(x_i^{\text{sc}}, y_i^{\text{sc}})\}_{i=1}^{N^{\text{sc}}}$ come from a single domain. If a natural grouping variable is available, such as an environment or a demographic attribute, it can be used to partition $\mathbf{D}^{\text{sc}}$ into multiple sources. Otherwise, each dataset $\mathbf{D}^{(k)}$ can be formed by random subsampling with replacement from $\mathbf{D}^{\text{sc}}$. This sub-sampling procedure can be justified when the source distribution is a mixture of heterogeneous subpopulations: repeated random sub-sampling with replacement increases the chance that some sub-samples approximate an individual subpopulation. Consequently, when the target distribution consists of the same subpopulations but with different mixing weights, treating the sub-samples as distinct sources enables us to develop procedures that are robust to changes in mixture proportions. This idea has been used in the maximin effect (Meinshausen & Bühlmann, 2015), and our approach builds on the same principle. Throughout the paper, even in the single-source setting, we treat each $\mathbf{D}^{(k)}$ as if it were drawn from a distinct source domain, thereby framing the problem as a multi-source domain adaptation task. For notational and modeling convenience, we refer to these sub-samples as "sources."

### 3.2 PROPOSED AMBIGUITY SET: HIGH-LEVEL FORMULATION

Let $\hat{P}_{Y|X}^{(k)}$ denote the estimated conditional distribution based on the $k$th source $\mathbf{D}^{(k)}$, and let $\hat{P}_X^{\text{tg}}$ be an estimator of the target input distribution. Details on how these estimators are constructed will be provided in Section 3.3. Given $\epsilon_1, \epsilon_2 \geq 0$ and $\bar{\beta} \in \Delta_{K-1}$, the proposed ambiguity set is defined as

$$\mathcal{Q} = \mathcal{Q}(\epsilon_1, \epsilon_2, \bar{\beta}) = \left\{ Q = (Q_X, Q_{Y|X}) \;\middle|\; \begin{array}{l} Q_{Y|X} = \sum_{k=1}^K \beta_k \hat{P}_{Y|X}^{(k)}, \quad \beta \in \Delta_{K-1}, \\ D_1\left(Q_X, \hat{P}_X^{\text{tg}}\right) \leq \epsilon_1, D_2\left(\beta, \bar{\beta}\right) \leq \epsilon_2 \end{array} \right\}, \quad (2)$$

where $\Delta_{K-1} = \{(\beta_1, \ldots, \beta_K) : \sum_{k=1}^K \beta_k = 1, \ \beta_k \geq 0\}$ denotes the $(K-1)$-dimensional simplex, and $D_1$ and $D_2$ are suitable divergence measures. Further details on the choices of $D_1$ and $D_2$ are provided in Section 3.3. The choices of $\hat{P}_{Y|X}^{(k)}$ and $D_1$ play a crucial role in ensuring the computational tractability of the proposed method.

The core idea behind the ambiguity set in (2) is that the target conditional distribution of the output given the input is expressed as a mixture of conditional distributions estimated from multiple source domains. To this end, we estimate different conditional distributions $\hat{P}_{Y|X}^{(k)}$ from sub-samples, and their mixtures form a sufficiently diverse class capable of containing the target conditional distribution. The radii $\epsilon_1$ and $\epsilon_2$ control the size of the ambiguity set and, consequently, the degree of distributional uncertainty allowed. The mixing vector $\beta$ is centered around a reference vector $\bar{\beta}$, which can encode prior knowledge about the relative importance of each source. If no prior information is available, one can set $\bar{\beta}$ in proportion to the sample sizes of the source groups. In our experiments, each group was constructed with the same number of samples. Thus, we simply set $\bar{\beta}$ to the uniform vector.

In addition to modeling the conditional distribution of $Y$ via mixtures, the ambiguity set (2) also accounts for uncertainty in the input distribution of the target domain through the radius parameter $\epsilon_1$. This additional layer of robustness is particularly beneficial when the number $N^{\text{tg}}$ of target samples is limited. In such cases, the empirical target input distribution $\hat{P}_X^{\text{tg}}$ may be a poor estimate of the true $P_X^{\text{tg}}$, and small perturbations in the input space can lead to large variations in model performance. By allowing the input distribution to vary within a controlled neighborhood, the method hedges against this sampling variability and prevents overfitting to a small or biased target sample. Such robustness is especially useful in practice, as limited or imbalanced target samples often fail to capture the diversity of the true domain.

The proposed ambiguity set in (2) is inspired by recent work on maximin effect estimation in regression settings (Meinshausen & Bühlmann, 2015; Guo, 2024; Wang et al., 2025). These studies consider DRO formulations where the ambiguity set consists of mixtures of conditional distributions from multiple source domains. In contrast to our classification setting, their analysis is primarily on regression, using negative explained variance or squared error loss as the performance criterion. This choice offers computational benefits; for example, in linear regression it leads to a closed-form solution for the optimization problem (see Theorem 1 in Meinshausen & Bühlmann (2015)). In classification settings, however, developing a computationally feasible DRO algorithm under an ambiguity set of the form (2) presents new challenges. Addressing these challenges and designing an efficient algorithm for this setting constitutes one of our main technical contributions, which we describe in the following subsections.

## 3.3 Detailed specification of the ambiguity set

To fully specify the ambiguity set $\mathcal{Q}$ in (2), we define the estimators $\hat{P}_X^{\text{tg}}$ and $\hat{P}_{Y|X}^{(k)}$ as well as the two divergence measures $D_1$ and $D_2$. The reference vector $\bar{\beta}$ is treated as fixed, while the radii $\epsilon_1$ and $\epsilon_2$ serve as tuning parameters that control the size of the ambiguity set.

We simply define $\hat{P}_X^{\text{tg}}$ as the empirical measure based on $\mathbf{D}^{\text{tg}}$. The construction of each $\hat{P}_{Y|X}^{(k)}$ depends on the available data. If sufficiently large datasets are available for each source, one can estimate $\hat{P}_{Y|X}^{(k)}$ separately using only the data from that source. However, in many practical settings, the amount of data per source may be limited. Moreover, when the distributions of different sources are similar, estimating them separately can be an inefficient use of data. To address this problem, in our experiments, we first train a single classification model using the entire source dataset. From this trained model, we extract a feature map $z : \mathcal{X} \to \mathcal{Z}$ by removing the final classification layer. Given the extracted features, we then estimate each $\hat{P}_{Y|X}^{(k)}$ (or the corresponding conditional density $\hat{p}_{Y|X}^{(k)}$) using a simple linear logistic regression trained independently on each source, treating the softmax outputs as practical probability estimates. This approach ensures that the proposed method remains efficient and scalable, even when the number of sources $K$ is large.

It is worth noting that the proposed method can be combined with existing UDA methods. For instance, after obtaining the feature map $z$, one may employ well-known approaches such as CDAN or STAR to construct $\hat{P}_{Y|X}^{(k)}$ using $z$ as the initial feature map, leading to further performance improvements. Further explanation and empirical results of this combination are provided in Section 4.2.

For the divergence $D_2$, we simply use the Euclidean distance. For $D_1$, we adopt the infinite-order Wasserstein distance, chosen for its computational tractability; see Section 3.4 for algorithmic de-

---

**Algorithm 1** Procedure for minimizing (3)

---

1: **Input:** step sizes $\eta_\theta, \eta_\beta, \eta_z$; initial values $\theta^{(0)}, \beta^{(0)}$
2: **repeat** for $t = 1, 2, \ldots$
3:     Sample $x_i^{\mathrm{tg}} \sim \hat{P}_X^{\mathrm{tg}}$ and compute $z_i^{\mathrm{tg}} = z(x_i^{\mathrm{tg}})$
4:     Compute $y^\circ(\beta^{(t-1)}, x_i^{\mathrm{tg}})$ as in (4)
                                                     ▷ Step 1: Update $z'$
5:     Update $z_i'$ using the gradient ascent and projection steps in (5)
                                                     ▷ Step 2: Update $\beta$
6:     Update $\beta^{(t)}$ using the exponentiated gradient ascent and projection steps in (6)
7:     Compute $y^\circ(\beta^{(t)}, x_i^{\mathrm{tg}})$
                                                     ▷ Step 3: Update $\theta$
8:     Update $\theta^{(t)}$ using the gradient descent in (7)
9: **until** convergence

---

tails. Recall that the Wasserstein distance of order $t \in [1, \infty)$ between two probability measures $P$ and $Q$ is defined as $W_t(Q, P) = \inf_\gamma \{ (\int c(x, x')^t \, d\gamma)^{1/t} \}$, where the infimum is taken over all couplings $\gamma$ of $Q$ and $P$, and $c(\cdot, \cdot)$ is a cost function. The infinite-order Wasserstein distance is then given by $W_\infty(Q, P) := \lim_{t \to \infty} W_t(Q, P)$.

In our case, we define the cost function as the Euclidean distance between feature representations, that is, $c(x, x') = \|z(x) - z(x')\|_2$, where $z$ denotes the feature map introduced earlier. This choice is often more effective than defining the cost function directly in the raw input space, particularly in high-dimensional settings (Ganin et al., 2016; Zeiler & Fergus, 2014; Krizhevsky et al., 2017).

## 3.4 LEARNING ALGORITHM

The proposed DRO estimator is obtained by solving the general DRO problem (1) with the ambiguity set defined in (2). In this subsection, we describe the detailed learning algorithm. We assume that $f^\theta(X)$ depends on $X$ only through the fixed representation $z(X)$ as defined in Section 3.3. Thus, we denote the model by $f_Z^\theta(z(X))$ in the remainder of this section. For notational convenience, we often treat the probability mass function $\hat{p}_{Y|X}^{(k)}(\cdot \mid X)$ as a probability vector. Before presenting the algorithm, we state the following proposition, which facilitates a computationally efficient implementation.

**Proposition 3.1** (Tractable surrogate reformulation). *Suppose that the feature map $z : \mathcal{X} \to \mathcal{Z}$ and the estimators $\hat{P}_{Y|X}^{(k)}$ are given. Let the divergence measures $D_1$ and $D_2$ be defined as in Section 3.3. For any $\epsilon_1, \epsilon_2 > 0$ and a fixed $\theta$, the maximization objective in the DRO problem (1) with the ambiguity set $\mathcal{Q}$ defined in (2), i.e., $\sup_{Q \in \mathcal{Q}} \mathbb{E}_Q[\ell(f^\theta(X), Y)]$ is upper-bounded by the following surrogate objective*

$$\sup_{\substack{\beta \in \Delta_{K-1}, \\ \|\beta - \bar{\beta}\|_2 \leq \epsilon_2}} \mathbb{E}_{\hat{P}_X^{\mathrm{tg}}} \left[ \sup_{\|z' - z(X)\|_2 \leq \epsilon_1} \ell\left(f_Z^\theta(z'), y^\circ(\beta, X)\right) \right], \tag{3}$$

*where*

$$y^\circ(\beta, x) := \sum_{k=1}^K \beta_k \, \hat{p}_{Y|X}^{(k)}(\cdot \mid x) \quad \text{for } x \in \mathcal{X} \tag{4}$$

*is the soft pseudo-label vector defined as a convex combination of source conditionals.*

This reformulation admits an efficiently computable algorithm that leverages adversarial feature perturbation $z'$ and soft pseudo-label $y^\circ$. The detailed rationale behind the approximation (3) is provided in Appendix A.1.

We now outline the procedure for minimizing (3). At a high level, the algorithm alternates between updating $z'$, $\beta$, and $\theta$ using a minimax optimization scheme implemented via stochastic gradient descent, as summarized in Algorithm 1. The detailed update rules are described below.

**1. Update of $z'$.** Suppose that $\theta$ and $\beta$ are fixed. Given a sample $x_i^{\text{tg}}$ from $\hat{P}_X^{\text{tg}}$, let $z_i^{\text{tg}} = z(x_i^{\text{tg}})$. To maximize the mapping $z' \mapsto \ell\left(f_Z^\theta(z'), y^\circ(\beta, x_i^{\text{tg}})\right)$, we perform a projected gradient ascent starting from $z_i^{\text{tg}}$:

$$
\begin{aligned}
z_i' &\leftarrow z_i^{\text{tg}} + \eta_z \nabla_z \ell\left(f_Z^\theta(z_i^{\text{tg}}), y^\circ(\beta, x_i^{\text{tg}})\right) \\
z_i' &\leftarrow \Pi_{\{z: \|z - z_i^{\text{tg}}\|_2 \leq \epsilon_1\}}(z_i'),
\end{aligned}
\tag{5}
$$

where $\nabla_z \ell(\cdot)$ denotes the gradient of $z \mapsto \ell(f_Z^\theta(z), y^\circ(\beta, x_i^{\text{tg}}))$, $\eta_z$ is the step size, and $\Pi_{\mathcal{A}}(\cdot)$ is the Euclidean projection onto the set $\mathcal{A}$.

**2. Update of $\beta$.** Given $\theta$ and $z_i'$, we update the mixture weights $\beta$ via a projected exponentiated gradient ascent (Sagawa et al., 2019):

$$
\begin{aligned}
\tilde{\beta}_k &\leftarrow \frac{\beta_k \exp\left(\eta_\beta \ell\left(f_Z^\theta(z_i'), \hat{p}_{Y|X}^{(k)}(\cdot \mid x_i^{\text{tg}})\right)\right)}{\sum_{j=1}^K \beta_j \exp\left(\eta_\beta \ell\left(f_Z^\theta(z_i'), \hat{p}_{Y|X}^{(j)}(\cdot \mid x_i^{\text{tg}})\right)\right)}, \quad \text{for } k = 1, \dots, K, \\
\beta &\leftarrow \Pi_{\{\beta: \|\beta - \bar{\beta}\|_2 \leq \epsilon_2\}}(\tilde{\beta}).
\end{aligned}
\tag{6}
$$

where $\eta_\beta$ is the step size. See Appendix A.2 for further details on this update.

**3. Update of $\theta$.** Given $z_i'$ and $\beta$, we update $\theta$ using a stochastic gradient descent:

$$
\theta \leftarrow \theta - \eta_\theta \nabla_\theta \ell\left(f_Z^\theta(z_i'), y^\circ(\beta, x_i^{\text{tg}})\right),
\tag{7}
$$

where $\eta_\theta$ is the step size.

Note that the role of $\beta$ is to form an adversarial mixture over these estimated conditionals. Specifically, the update rule in (6) assigns larger weights to conditional estimators that induce higher loss under the current classifier $f^\theta$. The model parameters $\theta$ are then updated in (7) to minimize the resulting adversarial objective. By iterating these updates, the classifier becomes robust to conditional uncertainty and potential mixture shifts. A detailed analysis of the stability of the learning algorithm is provided in Appendix A.3.

## 4 EXPERIMENTS

In this section, we evaluate the proposed method in two experimental settings using widely adopted benchmark datasets. The first experiment considers digit recognition tasks across three domains—MNIST, SVHN, and USPS—and compares the proposed method with existing UDA approaches, with particular emphasis on scenarios where the amount of target-domain data is limited. The second experiment examines the robustness of our method in the presence of spurious correlations, using popular benchmarks such as Waterbirds, CelebA, and Colored MNIST, where distribution shifts are induced by non-causal attributes. In the first experiment, one of MNIST, SVHN, or USPS is used as the single source domain, and one of the remaining two serves as the target domain, resulting in three source–target pairs. In the second experiment, each benchmark dataset consists of a single source domain and a single target domain as described below. Thus, all experiments are conducted under the single-source UDA setting. See Appendix A.5 for more detailed descriptions of the datasets.

**MNIST, SVHN, and USPS** (LeCun et al., 2002; Netzer et al., 2011; Hull, 2002): Three benchmark datasets for handwritten digit recognition from different domains. MNIST contains grayscale images of digits from 0 to 9, SVHN consists of color images of house numbers obtained from Google Street View, and USPS includes grayscale images of handwritten digits collected from postal envelopes. For each dataset, we randomly sampled $10^2$ and 10 unlabeled target samples per class.

**Waterbirds, CelebA, and CMNIST** (Sagawa et al. (2019); Liu et al. (2015); Arjovsky et al. (2019)): Three benchmark datasets widely used to study spurious correlations. Each dataset is divided into four groups, where spurious attributes (e.g., background in Waterbirds, gender in CelebA, or color in CMNIST) are correlated with labels. In all cases, three majority groups are combined to form the single-source domain, and the remaining minority group serves as the target domain. For Waterbirds, CelebA, and CMNIST, the unlabeled target sample sizes $N^{\text{tg}}$ are 56, 1,387, and 2,998, respectively.

| Method | SVHN → MNIST | | MNIST → USPS | | USPS → MNIST | |
|---|---|---|---|---|---|---|
| | $10^2$ | 10 | $10^2$ | 10 | $10^2$ | 10 |
| ERM (Src-only) | 59.6± 1.8 | | 63.4± 1.4 | | 60.4± 5.7 | |
| DANN | 66.0± 4.9 | 61.2± 1.8 | 82.0± 3.9 | 74.3± 5.7 | 74.8± 6.7 | 51.1± 5.0 |
| CDAN | 63.4±1.8 | 56.9±0.9 | 80.8±1.4 | 62.0±1.8 | 58.3±4.6 | 54.8±5.9 |
| MK-MMD | 50.0±3.0 | 48.3±1.0 | 63.3±1.1 | 41.3±5.5 | 57.6±3.2 | 32.1±3.9 |
| ATDOC | 83.3±9.1 | 59.1±5.7 | 91.1±0.8 | 73.6±4.2 | 92.6±1.3 | 87.0±3.6 |
| STAR | 76.4±1.5 | 66.8±0.9 | 90.3±1.9 | 81.3±7.6 | 94.5±0.7 | 85.2±2.7 |
| CORAL | 75.4±2.7 | 63.6±0.9 | 90.4±0.7 | 85.4±0.5 | 75.9±1.9 | 64.7±3.5 |
| MCD | 79.1±1.0 | 61.3±0.7 | 89.3±1.6 | 84.5±2.3 | 96.1±1.6 | 85.9±4.0 |
| Ours (ERM) | 92.0±1.6 | 87.0±0.9 | 92.1±1.1 | 87.1±3.6 | 90.3±2.2 | 86.4±2.5 |
| Ours (CDAN) | 92.5±2.3 | 87.0±2.0 | 93.5±1.3 | 87.8±1.5 | 91.5±1.9 | 87.0±1.9 |
| Ours (STAR) | **94.4**±1.7 | **91.3**±1.1 | **95.6**±1.0 | **91.2**±1.4 | **97.3**±0.8 | **93.0**±2.8 |
| *Ours (STAR) | 92.9±2.1 | 85.1±1.2 | 93.6±1.5 | 90.5±0.6 | 96.6±1.4 | 87.6±3.2 |

Table 1: Comparison of test accuracies across different target sample sizes for three domain adaptation benchmarks: SVHN, MNIST, and USPS. Each block reports results under two target data sizes: $10^2$ and 10 unlabeled samples per class. Here, Ours(·) denotes our methods built upon different base classifiers, and *Ours(·) denotes the versions tuned via LODO-CV. Boldface indicates the best performance.

## 4.1 EXPERIMENTAL SETUP

Following Meinshausen & Bühlmann (2015), in all our experiments we set $K = 10$ and draw each sub-sample of size $N = N^{\text{sc}}/5$ with replacement, which we denote by $\mathbf{D}^{(k)} = \{(x_i^{(k)}, y_i^{(k)})\}_{i=1}^N$, $k = 1, \ldots, 10$. A detailed analysis regarding the choice of $K$ is provided in Appendix A.4. We then use the entire source dataset to learn the feature map $z$. For constructing a base classifier $\hat{P}_{Y|X}^{(k)}$, we use three approaches: training a simple linear logistic regression model (ERM) on $\{(z(x_i^{(k)}), y_i^{(k)})\}_{i=1}^N$, and training CDAN and STAR, both initialized with the learned feature map $z$. For the backbone, we use two different models. In the first setting, we adopt a deep neural network, following the architecture in (Ganin et al., 2016). In the second setting, we employ ResNet-50 following (Sagawa et al., 2019).

For hyperparameter selection, we adopt a validation-based selection strategy to ensure a fair comparison across methods. Specifically, we assume the availability of a small labeled target validation set containing 10 samples per class, which is used to perform a grid search over $\epsilon_1 \in \{0, 0.2, 0.4, 0.6, 0.8, 1\}$ and $\epsilon_2 \in \{0, 0.2, 0.4, 0.6, 1\}$. This setting is consistent with those used in prior domain adaptation studies (Yue et al., 2023; Courty et al., 2017b; Saito et al., 2018), and the same selection procedure is applied to all competing methods. This choice allows us to report target-domain performance without being confounded by differences in hyperparameter selection strategies. In addition, we report the performance of our method using leave-one-domain-out cross validation (LODO-CV) (Gulrajani & Lopez-Paz, 2020), which does not rely on labeled target validation data and provides a complementary evaluation. Results obtained under this setting are marked with an asterisk (*Ours(·)). We compare our method with several baselines commonly used in domain adaptation and robust learning: DANN (Ganin et al., 2016), CDAN (Long et al., 2018), MK-MMD (Long et al., 2015), CORAL (Sun & Saenko, 2016), MCD (Saito et al., 2018), ATDOC (Tang et al., 2020), STAR (Lu et al., 2020), GroupDRO, ICON (Yue et al., 2023), DRUDA (Wang & Wang, 2024), and PDE (Deng et al., 2023).

## 4.2 RESULTS ON DIGIT RECOGNITION TASKS

In the first experiment, we consider two target sample sizes: $10^2$ and 10 per class. This setup allows us to evaluate whether our method can maintain robust performance even when only a very small number of target samples are available for training. Table 1 summarizes the results for three source–target domain pairs. Our method (Ours(·)) consistently achieves the best performance across all tasks and target sample sizes. Notably, the performance can be further improved by combining our approach with existing UDA methods such as CDAN and STAR. For example, when us-

| Method | Group label | Waterbirds Test Acc | CelebA Test Acc | CMNIST Test Acc |
|---|---|---|---|---|
| ERM (Src-only) | × | 48.4±0.9 | 35.5±0.6 | 0.9±0.5 |
| DANN | × | 35.8±4.5 | 23.5±2.1 | 0.9±1.8 |
| CDAN | × | 46.2±1.8 | 24.6±1.5 | 1.2±0.4 |
| MK-MMD | × | 45.1±1.2 | 27.7±2.9 | 2.8±1.3 |
| ATDOC | × | 47.3±1.4 | 31.8±1.4 | 3.1±0.9 |
| STAR | × | 49.8±5.6 | 24.4±2.4 | 2.2±2.7 |
| CORAL | × | 50.9±2.9 | 31.7±1.9 | 1.7±0.4 |
| MCD | × | 59.0±3.1 | 30.7±2.5 | 1.9±1.9 |
| DRST | × | 37.1±6.0 | 29.5±4.6 | 1.0±0.7 |
| ICON | × | 54.2±1.4 | 31.1±2.7 | 4.4±3.2 |
| GroupDRO | ✓ | 61.4±2.7 | 63.0±2.6 | 3.4±1.6 |
| GroupDRO (with Tgt) | ✓ | 90.6±0.2 | 89.3±1.3 | 73.1±0.3 |
| PDE | ✓ | 57.1±6.6 | 55.0±5.5 | 1.3±1.2 |
| Ours (ERM) | × | **87.3**±2.1 | **85.0**±4.1 | **7.5**±0.5 |
| *Ours (ERM) | × | 83.3±2.7 | 76.0±3.8 | 4.9±0.7 |

Table 2: Comparison of test accuracies across Waterbirds, CelebA, and CMNIST. ERM (Src-only) and GroupDRO are trained solely on source domain data, without using any target domain samples. GroupDRO (with Tgt) denotes the setting where both source labeled data and target labeled data are available during training. Boldface indicates the best performance except for models that use labeled target data for training.

ing CDAN as the base classifier, our method improves upon classical CDAN by $+29.1\%$ on the SVHN→MNIST task. Although *Ours(·) shows a moderate drop compared to Ours(·), it remains consistently competitive and still outperforms baselines that rely on labeled target data for tuning.

These improvements highlight the twofold strength of our framework. First, the ambiguity set explicitly accounts for uncertainty in the target input distribution through the $\epsilon_1$-radius. This design is particularly beneficial under extreme data scarcity, where the empirical distribution $\hat{P}_X^{\text{tg}}$ may be a poor approximation of the true $P_X^{\text{tg}}$. By allowing controlled perturbations, the model hedges against sampling variability and mitigates overfitting to small or biased target samples. Second, incorporating UDA methods such as CDAN or STAR enables the construction of conditional estimators that generalize better to the target domain. The formulation of the ambiguity set as a mixture of these refined conditionals produces an uncertainty set more closely aligned with the true target distribution. Consequently, the final model trained under this set achieves better generalization to target than mixtures based solely on ERM conditionals.

### 4.3 RESULTS ON SPURIOUS CORRELATION BENCHMARKS

Table 2 reports the test accuracies on the target domain across the three spurious-correlation benchmarks. Overall, most baseline methods show limited generalization to the target domain, with some performing even worse than standard ERM. This underscores the difficulty of these benchmarks, where spurious correlations and target-data scarcity occur simultaneously. It is well-known that classical UDA methods such as DANN, CDAN, and MK-MMD suffer from limited generalization when spurious correlations exist across domains, as they mainly match marginal input distributions without preserving semantic invariance (Johansson et al., 2019; Zhao et al., 2019). This often aligns spurious attributes (e.g., background, gender, or color) instead of causally related features, which can hurt generalization to the target domain. Pseudo-labeling methods such as STAR and ATDOC are also vulnerable, since spurious correlations can induce inaccurate pseudo-labels that propagate errors during training.

In contrast, our method does not rely on reducing the distance between marginal input distributions. Instead, it explicitly considers a set of plausible target distributions through a distributional ambiguity set and optimizes for the worst-case target risk within this set. By accounting for distributional variations that may arise from spurious attributes, this formulation guides the model to focus on predictive features that are invariant across the plausible target distributions, thereby enhancing robustness to spurious correlations. As a result, it significantly outperforms existing baselines, par-

ticularly in scenarios where spurious correlations dominate and unlabeled target data is extremely limited. Compared to ERM, it improves target domain accuracy by $+38.9\%$ on Waterbirds, $+49.5\%$ on CelebA, and $+6.6\%$ on CMNIST. Similar to the first experiment, *Ours($\cdot$) shows a slight performance reduction compared to Ours($\cdot$), yet still achieves stronger performance than all baselines.

## 4.4 SENSITIVITY ANALYSIS OF THE HYPERPARAMETERS $\epsilon_1$ AND $\epsilon_2$

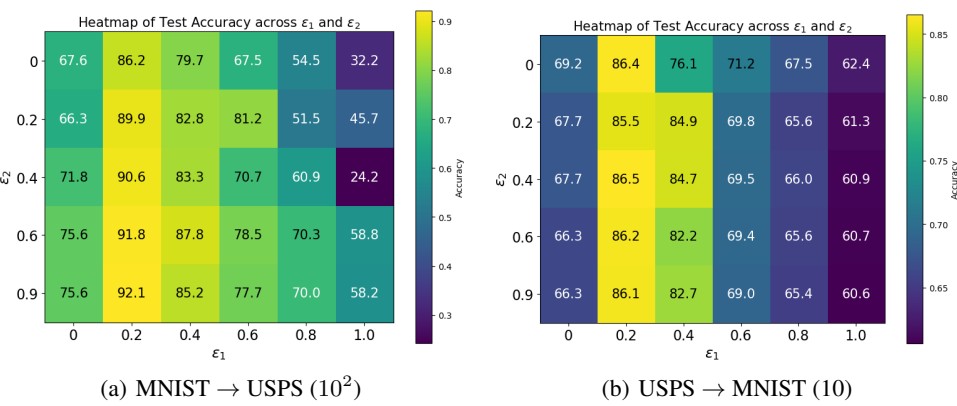

(a) MNIST $\rightarrow$ USPS ($10^2$)    (b) USPS $\rightarrow$ MNIST (10)

Figure 1: Heatmaps of average test accuracy across ($\epsilon_1$,$\epsilon_2$).

In this subsection, we analyze how the hyperparameters $\epsilon_1$ and $\epsilon_2$ affect model performance through a detailed sensitivity study. Figures 1 report heatmaps of the average test accuracy in different combinations of $\epsilon_1$ and $\epsilon_2$. Figure 1(a) corresponds to the MNIST$\rightarrow$USPS task with $10^2$ unlabeled target samples per class, while Figure 1(b) shows results for the USPS$\rightarrow$MNIST task with only 10 unlabeled target samples per class.

In Figure 1, the baseline setting $(\epsilon_1, \epsilon_2) = (0, 0)$ performs noticeably worse than configurations that introduce moderate uncertainty. Increasing either $\epsilon_1$ or $\epsilon_2$ generally improves accuracy, showing that controlled covariate or conditional perturbations enhance generalization. When the target sample size is moderately large, both hyperparameters exhibit a broad plateau of strong performance. As shown in Figure 1(a), accuracy remains high for $\epsilon_1 \in \{0.2, 0.4\}$ and $\epsilon_2 \geq 0.2$, demonstrating that the method is insensitive to small variations in these hyperparameters.

Under extreme target-data scarcity, the effect of $\epsilon_1$ becomes more prominent. Figure 1(b) shows that accuracy changes sharply along the $\epsilon_1$ axis, especially for $\epsilon_1 \geq 0.6$, while remaining relatively stable across values of $\epsilon_2$. This behavior reflects the setting in which $\hat{P}_X^{\text{tg}}$ is estimated from very few samples, making robustness to covariate-level variability particularly important.

The appropriate value of $\epsilon_1$ depends on the scale of the learned embedding, so no single universal choice applies across models. In practice, accuracy remains stable over a broad range of $\epsilon_1$ values (Figure 1), suggesting that approximate or heuristic selections work well in most cases. In contrast, $\epsilon_2$ can be set to a relatively large value (e.g., $\infty$) when no prior knowledge is available, as conditional mixing does not induce instability.

## 5 CONCLUSION

We presented a DRO framework for UDA that jointly models uncertainty in both the target covariate distribution and the conditional label distribution. By formulating an ambiguity set over mixtures of source conditionals and allowing controlled perturbations in target inputs, our method directly addresses two challenges that have gained increasing attention in recent research: the scarcity of unlabeled target data and the presence of spurious correlations in the source domain. Extensive experiments on digit recognition and spurious correlation benchmarks demonstrated consistent and substantial performance gains over strong baselines, highlighting the method's robustness and effectiveness. Although our experiments focus on vision benchmarks, extending the framework to other modalities such as NLP or time-series data is a natural next step.

ACKNOWLEDGMENTS

This work was supported by the National Research Foundation of Korea (NRF) grant funded by the Korea government (MSIT) (No. RS-2023-00240861, RS-2025-24523569, RS-2022-NR068758), a Korea Institute for Advancement of Technology (KIAT) grant funded by the Korea Government (MOTIE) (RS-2024-00409092, 2024 HRD Program for Industrial Innovation), and Institute of Information & Communications Technology Planning & Evaluation(IITP)-Global Data-X Leader HRD program grant funded by the Korea government (MSIT) (IITP-2024-RS-2024-00441244).

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

# A APPENDIX

## A.1 PROOF OF PROPOSITION 3.1

In this subsection, we provide the detailed rationale of the tractable reformulation stated in Proposition 3.1. With the ambiguity set $\mathcal{Q}$ defined in (2), the inner maximization in the DRO objective can be expressed as

$$\sup_{Q \in \mathcal{Q}} \mathbb{E}_Q[\ell\left(f^\theta(X), Y\right)] = \sup_{Q \in \mathcal{Q}} \mathbb{E}_{Q_X} \mathbb{E}_{Q_{Y|X}} [\ell\left(f^\theta(X), Y\right)] \tag{8}$$

$$= \sup_{\substack{\beta \in \Delta_{K-1}, \ W_\infty(Q_X, \hat{P}_X^{\mathrm{tg}}) \leq \epsilon_1 \\ \|\beta - \bar{\beta}\|_2 \leq \epsilon_2}} \mathbb{E}_{Q_X} \mathbb{E}_{Q_{Y|X}^\beta} \left[\ell\left(f^\theta(X), Y\right)\right], \tag{9}$$

where

$$Q_{Y|X}^\beta = \sum_{k=1}^K \beta_k \hat{P}_{Y|X}^{(k)}.$$

We recall the following lemma from Staib & Jegelka (2017), which will be useful in our reformulation.

**Lemma A.1.1** (*Staib & Jegelka (2017), Proposition 3.1*). *Suppose that the cost function $c(\cdot, \cdot)$ used to define the Wasserstein distance is a metric on $\mathcal{X}$. Let $P$ be a Borel probability measure on $\mathcal{X}$, and let $f : \mathcal{X} \to \mathbb{R}$ be a measurable function. Then, for any $\epsilon \geq 0$,*

$$\sup_{W_\infty(Q,P) \leq \epsilon} \mathbb{E}_Q[f(X)] = \mathbb{E}_P \left[ \sup_{x' \in B_\epsilon(X)} f(x') \right],$$

*where $B_\epsilon(x) = \{x' \in \mathcal{X} : c(x, x') \leq \epsilon\}$ and $\mathbb{E}_P$ denotes the expectation under $X \sim P$.*

By Lemma A.1.1, we have

$$\sup_{W_\infty(Q_X, \hat{P}_X^{\mathrm{tg}}) \leq \epsilon_1} \mathbb{E}_{Q_X} \left[ \mathbb{E}_{Q_{Y|X}^\beta} \left[ \ell\left(f^\theta(X), Y\right) \right] \right]$$

$$= \mathbb{E}_{\hat{P}_X^{\mathrm{tg}}} \left[ \sup_{x : \|z(x) - z(X)\|_2 \leq \epsilon_1} \mathbb{E}_{Q_{Y|X}^\beta} \left[ \ell\left(f^\theta(x), Y\right) \right] \right]$$

$$= \mathbb{E}_{\hat{P}_X^{\mathrm{tg}}} \left[ \sup_{x : \|z(x) - z(X)\|_2 \leq \epsilon_1} \mathbb{E}_{Q_{Y|X}^\beta} \left[ \ell\left(f_Z^\theta(z(x)), Y\right) \right] \right]$$

$$\leq \mathbb{E}_{\hat{P}_X^{\mathrm{tg}}} \left[ \sup_{z' : \|z' - z(X)\|_2 \leq \epsilon_1} \mathbb{E}_{Q_{Y|X}^\beta} \left[ \ell\left(f_Z^\theta(z'), Y\right) \right] \right]$$

Therefore, (8) is upper-bounded by

$$\sup_{\substack{\beta \in \Delta_{K-1}, \\ \|\beta - \bar{\beta}\|_2 \leq \epsilon_2}} \mathbb{E}_{\hat{P}_X^{\mathrm{tg}}} \left[ \sup_{\|z' - z(X)\|_2 \leq \epsilon_1} \mathbb{E}_{Q_{Y|X}^\beta} \left[ \ell\left(f_Z^\theta(z'), Y\right) \right] \right].$$

By treating the probability mass function $\hat{p}_{Y|X}^{(k)}(\cdot \mid X)$ as a probability vector, we define the soft pseudo-label vector as

$$y^\circ(\beta, x) := \sum_{k=1}^K \beta_k \, \hat{p}_{Y|X}^{(k)}(\cdot \mid x). \tag{10}$$

Recall that the cross-entropy loss is defined by

$$\ell\left(f_Z^\theta(z'), Y\right) = -\sum_{y \in \mathcal{Y}} \mathbb{I}[Y = y] \log \pi_Z^\theta(y \mid z'),$$

where $\pi_Z^\theta(\cdot \mid z')$ denotes the predicted class probability vector obtained by applying the softmax function to $f_Z^\theta(z')$ and $\mathbb{I}[\cdot]$ is the indicator function. Therefore, we have

$$
\begin{aligned}
\mathbb{E}_{Q_{Y|X=x}^\beta} \left[ \ell \left( f_Z^\theta(z'), Y \right) \right] &= \mathbb{E}_{Q_{Y|X=x}^\beta} \left[ -\sum_{y \in \mathcal{Y}} \mathbb{I}[Y = y] \log \pi_Z^\theta(y \mid z') \right] \\
&= -\sum_{y \in \mathcal{Y}} \left( \sum_{k=1}^K \beta_k \, \hat{p}_{Y|X}^{(k)}(y \mid x) \right) \log \pi_Z^\theta(y \mid z') \\
&= \ell \left( f_Z^\theta(z'), y^\circ(\beta, x) \right).
\end{aligned}
\tag{11}
$$

Here, we slightly abuse the notation for $\ell(\cdot, \cdot)$, interpreting $\ell(f_Z^\theta(z'), y^\circ(\beta, x))$ as the cross-entropy between the two probability vectors $\pi_Z^\theta(\cdot \mid z')$ and $y^\circ(\beta, x)$.

Combining the above, we obtain the following surrogate objective for (8):

$$
\sup_{\substack{\beta \in \Delta_{K-1}, \\ \|\beta - \bar\beta\|_2 \leq \epsilon_2}} \mathbb{E}_{\hat{P}_X^{\mathrm{tg}}} \left[ \sup_{\|z' - z(X)\|_2 \leq \epsilon_1} \ell \left( f_Z^\theta(z'), y^\circ(\beta, X) \right) \right].
\tag{12}
$$

### A.1.1 ON THE TIGHTNESS OF THE RELAXATION GAP OF THE SURROGATE LOSS

Our surrogate objective (12) can be strictly smaller than the original objective. The gap arises from the inequality

$$
\sup_{x: \|z(x) - z(X)\|_2 \leq \epsilon_1} \mathbb{E}_{Q_{Y|X}^\beta} \left[ \ell \left( f_Z^\theta(z(x)), Y \right) \right] \leq \sup_{z': \|z' - z(X)\|_2 \leq \epsilon_1} \mathbb{E}_{Q_{Y|X}^\beta} \left[ \ell \left( f_Z^\theta(z'), Y \right) \right].
$$

This inequality can be strict when the feature map $z(\cdot)$ does not cover the entire $\epsilon_1$-ball. In many practical settings, however, embedding distributions produced by deep networks are modeled under the assumption that the pushforward distribution of $X$ through $z(\cdot)$ admits a positive Lebesgue density on, or in a neighborhood of, its effective support.

Raw images often lie near a low-dimensional manifold in the input space and may not possess a Lebesgue density. After several nonlinear transformations, internal representations typically exhibit different geometric behavior and are commonly treated as "thickened" manifolds with nonzero volume in the ambient space. The widespread use of the Fréchet Inception Distance (FID) illustrates this modeling perspective, as the metric implicitly assumes a non-degenerate covariance structure for deep embeddings.

Under the above modeling assumption, the image of $z(\cdot)$ generally overlaps substantially with the $\epsilon_1$-ball, making the relaxation introduced after Lemma A.1.1 nearly tight and the resulting upper bound a close approximation of the original hierarchical objective.

### A.2 EXPONENTIATED GRADIENT ASCENT FOR UPDATING $\beta$

We first review the exponentiated gradient method briefly and then describe its application to our $\beta$-update procedure.

**Exponentiated gradient ascent method** The exponentiated gradient ascent method can be interpreted as a mirror ascent step over the simplex, obtained by maximizing the first-order approximation of the objective with a KL divergence regularization. Specifically, let $\mathcal{L}(\beta)$ be a differentiable objective function. Then, the exponentiated gradient ascent update from $\beta^{(t)}$ is given by

$$
\beta^{(t+1)} = \arg \max_{\beta \in \Delta_{K-1}} \left\{ \mathcal{L}(\beta^{(t)}) + \nabla_\beta \mathcal{L}(\beta^{(t)})^\top (\beta - \beta^{(t)}) - \frac{1}{\eta_\beta} D_{\mathrm{KL}} \left( \beta \,\|\, \beta^{(t)} \right) \right\},
$$

where $\nabla_\beta$ denotes the gradient of the mapping $\beta \mapsto \mathcal{L}(\beta)$, and $\eta_\beta > 0$ is the step size. Since the terms involving $\beta^{(t)}$ are constant with respect to $\beta$, this is equivalent to

$$
\beta^{(t+1)} = \arg \max_{\beta \in \Delta_{K-1}} \left\{ \nabla_\beta \mathcal{L}(\beta^{(t)})^\top \beta - \frac{1}{\eta_\beta} D_{\mathrm{KL}} \left( \beta \,\|\, \beta^{(t)} \right) \right\}.
$$

By the KKT optimality conditions, the maximization above admits the following closed-form solution:

$$\beta_k^{(t+1)} = \frac{\beta_k^{(t)} \exp\left(\eta_\beta \left[\nabla_\beta \mathcal{L}(\beta^{(t)})\right]_k\right)}{\sum_{j=1}^K \beta_j^{(t)} \exp\left(\eta_\beta \left[\nabla_\beta \mathcal{L}(\beta^{(t)})\right]_j\right)}, \qquad k = 1, \dots, K, \tag{13}$$

where $[A]_i$ denotes $i$-th component of $A$. See Kivinen & Warmuth (1997) for details.

**Detailed procedure for updating $\beta$** We now revisit our optimization problem in (3). Since the constraint $\|\beta - \bar{\beta}\|_2 \le \epsilon_2$ is handled by projection after the ascent step, we focus only on updating $\beta$ over the simplex $\Delta_{K-1}$. Suppose that $x_i^{\mathrm{tg}}$ and $z_i'$ are given. Then, we need to solve

$$\sup_{\beta \in \Delta_{K-1}} \left[\ell\left(f_Z^\theta(z_i'), y^\circ(\beta, x_i^{\mathrm{tg}})\right)\right].$$

This problem can be expressed as:

$$\begin{aligned}
\sup_{\beta \in \Delta_{K-1}} \ell\left(f_Z^\theta(z_i'), y^\circ(\beta, x_i^{\mathrm{tg}})\right) &= \sup_{\beta \in \Delta_{K-1}} \mathbb{E}_{Q_{Y|X=x_i^{\mathrm{tg}}}^\beta}\left[\ell\left(f_Z^\theta(z_i'), Y\right)\right] \\
&= \sup_{\beta \in \Delta_{K-1}} \sum_{k=1}^K \beta_k \mathbb{E}_{\hat{P}_{Y|X=x_i^{\mathrm{tg}}}^{(k)}}\left[\ell\left(f_Z^\theta(z_i'), Y\right)\right] \\
&= \sup_{\beta \in \Delta_{K-1}} \sum_{k=1}^K \beta_k \ell\left(f_Z^\theta(z_i'), \hat{p}_{Y|X}^{(k)}(\cdot \mid x_i^{\mathrm{tg}})\right),
\end{aligned}$$

where this expression follows from (4) and (11).

Setting

$$\mathcal{L}(\beta) := \sum_{k=1}^K \beta_k \ell\left(f_Z^\theta(z_i'), \hat{p}_{Y|X}^{(k)}(\cdot \mid x_i^{\mathrm{tg}})\right),$$

and applying the closed-form solution in (13), we obtain the following update rule:

$$\beta_k^{(t+1)} \leftarrow \frac{\beta_k^{(t)} \exp\left(\eta_\beta \ell\left(f_Z^\theta(z_i'), \hat{p}_{Y|X}^{(k)}(\cdot \mid x_i^{\mathrm{tg}})\right)\right)}{\sum_{j=1}^K \beta_j^{(t)} \exp\left(\eta_\beta \ell\left(f_Z^\theta(z_i'), \hat{p}_{Y|X}^{(j)}(\cdot \mid x_i^{\mathrm{tg}})\right)\right)}, \quad k = 1, \dots, K.$$

### A.3 EMPIRICAL ANALYSIS OF THE STABILITY OF LEARNING ALGORITHM

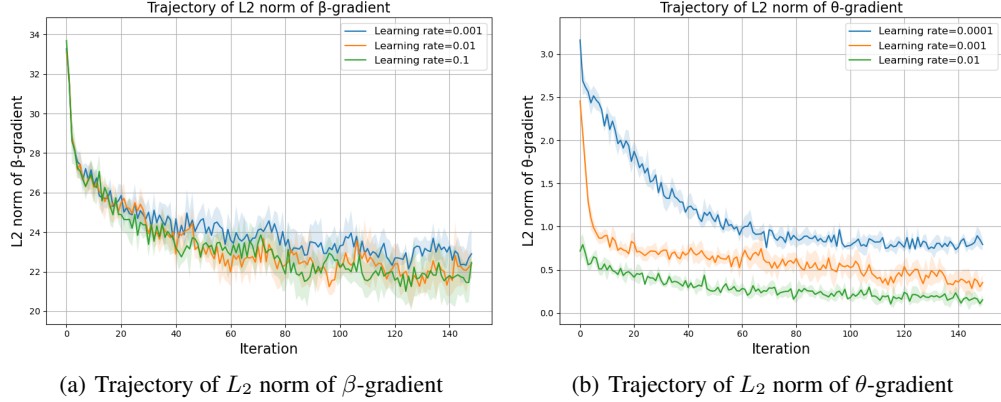

(a) Trajectory of $L_2$ norm of $\beta$-gradient   (b) Trajectory of $L_2$ norm of $\theta$-gradient

Figure 2: Gradient behavior of $\beta$ and $\theta$ under joint optimization.

In this subsection, we analyze the stability of our learning algorithm presented in Section 3.4. To assess this stability, we further examine the gradient behavior of the joint optimization procedure

by conducting an additional controlled experiment. Specifically, we empirically track the gradient trajectories of both the $\beta$-gradient (adversarial weights) and the $\theta$-gradient (model parameters).

Specifically, we fixed the hyperparameters $(\epsilon_1, \epsilon_2) = (0.6, 0.4)$ and performed a domain adaptation experiment on the SVHN→MNIST task using a mini-batch size of 128. At each iteration, we computed the $L_2$ norm of the gradients with respect to $\beta$ and $\theta$. For panel (a), the learning rate for $\theta$ was fixed at $10^{-3}$ while varying the learning rate for $\beta$. For panel (b), the learning rate for $\beta$ was fixed at $10^{-2}$ while varying the learning rate for $\theta$. For each learning rate in $\{10^{-1}, 10^{-2}, 10^{-3}\}$, we repeated the procedure 20 times to assess stability and convergence across different update magnitudes.

Figure 2 demonstrates the resulting gradient trajectories of $\beta$ and $\theta$, respectively. The main observations are summarized below.

- Across all learning rates, both $\beta$ and $\theta$ show monotonically decreasing and smooth gradient norms, indicating that the updates consistently move toward a stable point without sharp fluctuations.
- Even under the most aggressive learning rate ($10^{-1}$), the gradient norms remain well-controlled and do not exhibit any signs of gradient explosion, oscillation, or unstable plateauing, which are typical indicators of optimization instability in min–max procedures.
- The variance bands across 20 random repetitions are uniformly narrow, suggesting that the optimization dynamics are highly reproducible and not sensitive to stochastic variation in initialization or minibatch sampling.

These empirical findings demonstrate that the proposed joint optimization procedure is numerically stable, robust to learning-rate variation, and reliable across repeated runs.

The optimization structure follows the standard formulation of GroupDRO (Sagawa et al., 2019), whose convergence behavior has been analyzed in prior work. Similar min–max update rules have also been used in recent studies (Krueger et al., 2021; Zhou et al., 2021), indicating that this class of optimization procedures is generally well-behaved under practical settings.

## A.4 SELECTION OF PSEUDO-GROUPS $K$

In this subsection, we analyze practical strategies for selecting the number of pseudo-groups $K$ and conduct a sensitivity analysis. Selecting the number of pseudo-groups $K$ is critical in the single-source setting due to an inherent trade-off. If $K$ is too small, the diverse conditional structures in the source distribution may not be fully captured, limiting the representation of target conditional uncertainty. Conversely, increasing $K$ substantially raises computational and memory costs, as each pseudo-group requires training a separate conditional estimator $\hat{P}_{Y|X}^{(k)}$. Therefore, choosing an intermediate value of $K$ is essential. We discuss two practical strategies for selecting $K$ and analyze the sensitivity of the model to this choice.

The first strategy utilizes a small labeled target validation set to guide the choice of $K$. In our experiments, we used 10 labeled samples per class. Figure 7(a) visualizes the validation accuracy for the MNIST→USPS task. Accuracy increases steadily as $K$ grows from 2 to 7 across all $\epsilon_2$. Beyond this point, performance stabilizes: when $K \geq 7$ and $\epsilon_2 > 0.4$, accuracy reaches a plateau between 89% and 91%. Although the highest accuracy is observed at $K = 14$ with $\epsilon_2 = 1$, $K = 10$ serves as a practical compromise, balancing performance with computational efficiency.

Alternatively, $K$ can be selected via cross-validation on the source data, leveraging the Maximin effect (Meinshausen & Bühlmann, 2015). For each candidate $K$, the source data is partitioned into training and validation subsets. $K$ pseudo-groups are formed from the training subset to train conditional estimators $\hat{P}_{Y|X}^{(k)}$, which are then evaluated on the validation subset. The $K$ yielding the highest average score is selected. This provides a data-driven criterion independent of target labels. Figure 7(b) shows the validation accuracy obtained from 5-fold cross validation. one can see that validation performance improves with $K$ and stabilizes for $K \geq 9$ with $\epsilon_2 \in [0.4, 0.8]$. These results suggest that selecting $K$ in the range of 9 to 14 is appropriate.

To assess robustness, Figure 7(c) summarizes test accuracy across varying $K$ and $\epsilon_2$. Performance varies smoothly across different $\epsilon_2$, indicating low sensitivity to this parameter. Accuracy improves

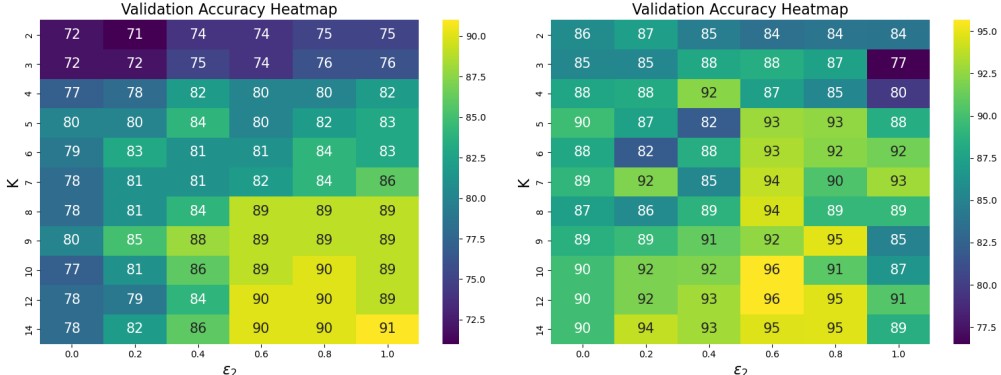

(a) Validation with a small labeled target set    (b) Validation using cross-validation

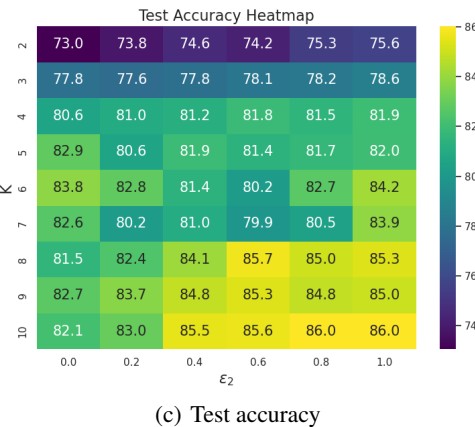

(c) Test accuracy

Figure 3: Heatmaps of average accuracy for the MNIST→USPS task with $10^2$ unlabeled target samples per class, shown across different values of $K$ and $\epsilon_2$. Panels (a) and (b) show validation accuracy obtained using a small labeled target set and cross-validation, respectively. Panel (c) shows the corresponding test accuracy. All results are averaged over 5 independent runs.

up to $K = 7$, after which it plateaus within a high-performing range (84%–86% for $K \geq 8$). Notably, both selection strategies converge to a consistent range of $K \in [9, 12]$. While larger values of $K$ may offer marginal gains, $K = 10$ is adopted as the default setting to balance accuracy with computational cost, which increases linearly with $K$.

## A.5 EXPERIMENTS DETAILS

In this section, we provide details regarding the datasets and baseline configurations. The source code is available at `https://github.com/kshwi5500/DRL_for_UDA`.

### A.5.1 DIGIT BENCHMARKS

**MNIST** (LeCun et al., 2002): A widely used benchmark dataset of handwritten digit recognition, consisting of grayscale images of digits from 0 to 9. Each image is of size $28 \times 28$ pixels, yielding 784-dimensional input vectors when flattened. The dataset contains a total of 70,000 images, with 60,000 images allocated for training and 10,000 images reserved for testing. The digit classes are balanced across 10 categories, corresponding to the labels 0 through 9. Due to its simplicity and accessibility, MNIST has been extensively used for evaluating classification algorithms, representation learning methods, and as a canonical testbed for new machine learning models.

**SVHN** (Netzer et al., 2011): The Street View House Numbers dataset, designed for digit recognition in natural scenes. It consists of cropped color images of digits ($32 \times 32$ pixels) obtained from Google Street View house numbers. SVHN provides 73,257 training images, 26,032 test images, and an additional set of 531,131 extra training images to facilitate large-scale training. The dataset covers 10 digit classes (0–9) and presents greater variability than MNIST due to cluttered backgrounds, varying illumination, and diverse font styles, making it a challenging and widely used benchmark in digit classification and domain adaptation studies.

**USPS** (Hull, 2002): The U.S. Postal Service (USPS) dataset is a benchmark collection of handwritten digits originally scanned from postal envelopes. Each image is a grayscale $16 \times 16$ pixel representation of digits ranging from 0 to 9, resulting in compact 256-dimensional input vectors when flattened. The dataset provides 7,291 training images and 2,007 test images, spanning 10 classes that are approximately balanced across digits. Compared to MNIST, USPS offers lower-resolution images and displays noticeable stylistic variations in handwriting, including slant, thickness, and shape differences, which make recognition more challenging. Due to these characteristics, USPS is often used alongside MNIST and SVHN as a complementary benchmark in digit classification tasks and as a target or source domain in domain adaptation studies.

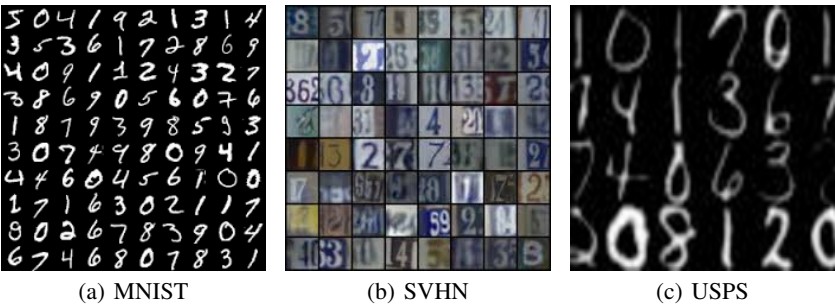

(a) MNIST          (b) SVHN          (c) USPS

Figure 4: Example images from the digit dataset.

### A.5.2 SPURIOUS CORRELATION BENCHMARKS

**Waterbirds** (Sagawa et al. (2019)): The Waterbirds dataset is designed to study the impact of spurious correlations in image classification. It is constructed by overlaying bird images from the CUB-200-2011 dataset (Wah et al. (2011)) onto background scenes from the Places dataset (Zhou et al. (2017)). This process induces a strong but spurious correlation between bird type and background. Specifically, the dataset is divided into four groups based on bird type (landbird vs. waterbird) and background (land vs. water): $g_1 = \{$landbird, land$\}$, $g_2 = \{$landbird, water$\}$, $g_3 = \{$waterbird, land$\}$, and $g_4 = \{$waterbird, water$\}$. Most samples belong to the aligned groups ($g_1$ and $g_4$), while the misaligned groups ($g_2$ and $g_3$) are relatively rare. This imbalance creates minority groups that highlight the difficulty of learning invariant predictors under spurious correlations. In our experiments, we follow prior work by treating the three majority groups as the single-source domain and

the minority group ($g_3$, waterbirds with a land background) as the target domain. The training set contains 4,739 source images and 56 unlabeled target images, while the target test set includes 642 samples, providing a controlled evaluation of out-of-distribution generalization.

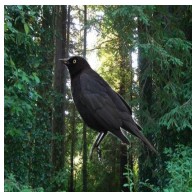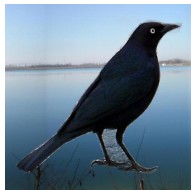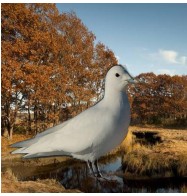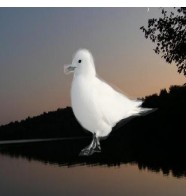

(a) Landbirds, Land (b) Landbirds, Water (c) Waterbirds, Land (d) Waterbirds, Water

Figure 5: Example images from the Waterbirds dataset.

**CelebA** (Liu et al. (2015)): The CelebA dataset contains large-scale celebrity face images annotated with multiple binary attributes. We consider the task of classifying blond vs. non-blond hair, where gender acts as a spurious attribute correlated with hair color. This naturally induces four groups based on hair color (blond vs. non-blond) and gender (male vs. female): $g_1 = \{$non-blond, female$\}$, $g_2 = \{$non-blond, male$\}$, $g_3 = \{$blond, female$\}$, and $g_4 = \{$blond, male$\}$. Most samples belong to $g_1$, $g_2$, and $g_3$, while the minority group $g_4$ (blond-haired males) is underrepresented, reflecting the spurious correlation between hair color and gender. The three majority groups are combined to form the single-source domain, and the minority group is treated as the target domain. The dataset provides 161,383 labeled source training images, 1,387 unlabeled target training images, and 642 target test images for evaluation.

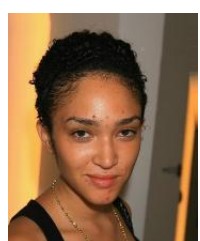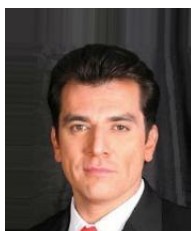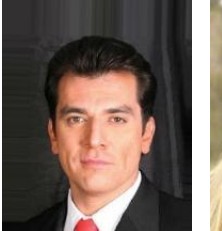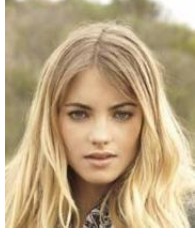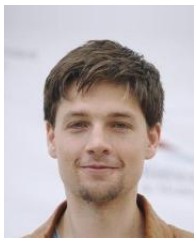

(a) Non-blond, Female (b) Non-blond, Male (c) Blond, Female (d) Blond, Male

Figure 6: Example images from the CelebA dataset.

**Colored MNIST (CMNIST)** (Arjovsky et al. (2019)): The Colored MNIST dataset is a synthetic variant of MNIST designed to study the effect of spurious correlations. We consider a binary classification task using digits 0 and 1, which yields four groups based on digit identity and color: $g_1 = \{$digit 0, red$\}$, $g_2 = \{$digit 0, green$\}$, $g_3 = \{$digit 1, red$\}$, and $g_4 = \{$digit 1, green$\}$. Most samples belong to the aligned groups ($g_1$, $g_4$, and $g_3$), while the minority group $g_2$ (digit 0 with green color) is underrepresented, capturing the spurious correlation between digit and color. In our experiments, the three majority groups are combined to form the single-source domain, and the minority group is used as the target domain. The dataset provides 26,002 labeled source training images, 2,998 unlabeled target training images, and 8,966 target test images.

### A.5.3 BASELINE DETAILS

- ERM (Src-only) The standard empirical risk minimization approach that optimizes average accuracy on the source training set. While simple and widely used, ERM does not incorporate any robust objective and cannot exploit unlabeled target data, making it insufficient under domain shift.

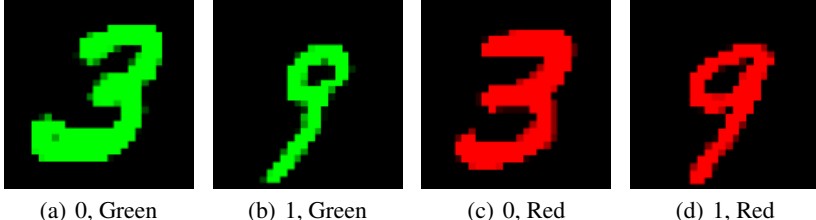

| (a) 0, Green | (b) 1, Green | (c) 0, Red | (d) 1, Red |

Figure 7: Example images from the CMNIST dataset.

- CDAN (Long et al., 2018): Extends DANN by conditioning the domain discriminator on both feature representations and classifier predictions. This coupling captures multimodal structures in the feature space, leading to more effective domain alignment.

- MK-MMD (Long et al., 2015): Minimizes the maximum mean discrepancy (MMD) across multiple kernels between source and target feature distributions. By leveraging multiple kernels, MK-MMD adapts flexibly to complex distribution shifts.

- CORAL (Sun & Saenko, 2016): Matches the second-order statistics (covariances) of source and target features to reduce domain discrepancy. This moment-matching approach is simple, efficient, and widely adopted in UDA.

- MCD (Saito et al., 2018): Employs two classifiers with a shared feature extractor. By maximizing their prediction discrepancy on target samples and then minimizing it, MCD encourages the extractor to learn target-discriminative features.

- ATDOC (Tang et al., 2020): Adapts classifiers by leveraging target-domain pseudo-labels in an adversarial manner. It progressively refines pseudo-labels to improve alignment and robustness in the absence of ground-truth target labels.

- STAR (Lu et al., 2020): Introduces stochastic classifiers to model diverse decision boundaries. By averaging predictions across multiple classifiers, STAR improves stability and robustness under domain shifts.

- Group DRO (Sagawa et al., 2019): A robust optimization method that minimizes the worst-case loss across predefined groups. Since it relies on explicit group labels in the training data, Group DRO cannot leverage unlabeled target data, which limits its applicability in unsupervised domain adaptation.

- Group DRO (with Tgt) (Sagawa et al., 2019): A robust optimization method that minimizes the worst-case loss across predefined groups. In this variant, we assume access to group labels from both the source and target domains during training. This setup is not realistic in unsupervised domain adaptation, since target group labels are unavailable in practice.

- ICON (Yue et al., 2023): Focuses on unsupervised domain adaptation by enforcing invariant consistency across source and target predictions. ICON leverages consistency regularization to learn representations robust to distributional shifts, achieving state-of-the-art performance in UDA.

- PDE (Deng et al., 2023): A method designed to combat spurious correlations by progressively expanding the training data from easy samples to harder ones. It identifies samples where the model relies on invariant features rather than spurious ones, effectively improving generalization on out-of-distribution data without requiring full group labels for all samples.

- DRUDA (Wang & Wang, 2024): A distributionally robust unsupervised domain adaptation (UDA) framework that addresses the distribution shift between source and target domains. By employing a distributionally robust optimization approach, it minimizes the worst-case risk over an uncertainty set around the target distribution, thereby enhancing the model's stability and performance in unlabeled target environments.

### A.6 DISCLOSURE ON LLM USAGE

We used large language models solely to aid in polishing the writing of this paper.

