# OpenReview forum: "Distributionally Robust Classification for Multi-source Unsupervised Domain Adaptation"
_ICLR.cc/2026/Conference — ICLR 2026 Poster_

### Official Review · Reviewer_fB8c · 2025-10-20

**Soundness:** 3
**Presentation:** 4
**Contribution:** 3
**Rating:** 6
**Confidence:** 3

**Summary:**

The paper introduces a novel method for unsupervised domain adaption based on the framework of distributional robust optimization. The key idea is to represent the conditiona distribution $P(Y|X)$ as the mixture of empiricial conditional distribution $\hat{P}^{(k)}(Y|X)$ from multiple sources and allows the pertubation of target input distribution over move inside a small Wasserstein ball. This method is plug-and-play, can be effectively integrated with existing UDA methods. Experiments on digits (MNIST/SVHN/USPS) and spurious-correlation suites (Waterbirds, CelebA, Colored-MNIST) show consistent gains and strong performance when target data are very limited.

**Strengths:**

1. The paper is quite well-written.

2. The proposed  method is quite intuitive and easy to understand, with tractable surrogate and a relatively simple algorithm.

3. The algorithm integrates seamlessly with existing UDA frameworks since its neat design that the input of the algorithm is the feature mapping $z$.

4. The empirical effectivness is validated on real-world datasets with non-trivial improvements.

**Weaknesses:**

Currently, the largest potential issue is the lack of discussion of the grouping strategy. Specifically, it is not clear for a single soruce distribution, how do we know the number of pseudo-sources $K$? The choice of $K$ should directly influence the performance given the main idea that the target conditional distribution is the mixture of empirical conditional distribution. It will be appreicated to provide a more detailed analysis regarding the choice of $K$ (e.g., heuristics, validation criteria, stability checks), plus a short sensitivity or ablation study.

I’m not an expert in UDA, so I’m unsure whether the current baselines reflect the latest methods. A quick check search leads me to two recent papers [1], [2]. Could you comment on their relevance and, if appropriate, explain why they weren’t included (e.g., different setting, data requirements, or incompatibility)? I’m completely open to your explanation.

Another potential issue the optimization objective is surrogate-based, i.e., we are always apporacing a suboptimal result. In this case, an analysis regarding the tightness of the gap will be appreciated. However, I totally understand if there have not been any since this is not the contribution of the paper. I will not change my assessment of the paper regardless of the absent of such analysis.

[1] Partial Identifiability for Domain Adaptation.

[2] Subspace identification for multi-source domain adaptation.

**Questions:**

Most questions have been proposed in the Weakness section.

A minor question:

During the analysis of impact of radius $(\epsilon_1, \epsilon_2)$, I do not quite understand the content from lines 1004-1007. The aurthors argue that $\epsilon_2$ will play a critical role when the target data is quite scarce. However, according to the Figure 5(b), the $\epsilon_2$ provides nearly no influence for a fixed $\epsilon_1$. Can the authors elaborate more on their arugments, or am I missing anything?

---

> ### Author Response · Authors · 2025-11-20
>
> We thank the reviewer for the careful reading of our paper. Below we provide a point-by-point response, and all corresponding revisions have been incorporated into the revised paper and highlighted in blue.
>
> ---
>
> **W1**
> >Detailed analysis regarding the choice of $K$ in the single-source setting
>
> We thank the reviewer for raising this important point. As the reviewer noted, selecting the number of pseudo-groups $K$ is critical in the single-source setting. There exists an inherent trade-off: if $K$ is too small, we cannot capture the diverse conditional structures in the source distribution, limiting the representation of target conditional uncertainty. On the other hand, increasing $K$ substantially raises computational and memory cost, since each pseudo-group requires training a separate conditional estimator $\hat{P}^{(k)}_{Y|X}$. For this reason, choosing an intermediate value of $K$ is essential. In the following, we introduce two practical strategies for selecting $K$ and present a sensitivity analysis based on test accuracy to assess the robustness of these choices.
>
> - **Selection of $K$ with a predefined validation set**: The first strategy is to use a predefined validation set to guide the choice of $K$. In our paper, we employ a small labeled target validation set consisting of 10 labeled samples per class.
> Figure 7 (https://drive.google.com/file/d/1Ngunn6kvw81VJr2-juE_LHyRRF59IXN_/view?usp=drive_link) in revision paper visualizes validation accuracy for the MNIST$\rightarrow$USPS task with $10^2$ unlabeled target samples per class. In Figure 7(a), validation accuracy increases steadily as $K$ grows from 2 to 7 across all $\epsilon_2$. Beyond this point, performance stabilizes: when $K \geq 7$ combined with $\epsilon > 0.4$, all achieve high validation accuracy between 89\% and 91\%, forming a clear plateau. Although the highest validation accuracy is observed at $K=14$, $\epsilon_2 = 1$ with 91\%, we selected $K=10$ as a practical compromise, balancing this performance with computational considerations.
>
> - **Selection of $K$ with cross validation**: Another practical strategy for selecting $K$ is to use cross validation, as suggested in the Maximin effect [1]. For each candidate $K$, we repeatedly partition the source data into training and validation subsets. Using the training subset, we form $K$ pseudo-groups to train $K$ conditional estimators ${\hat{P}^{(k)}_{Y|X}}$, and evaluate the model on the held-out validation subset. We repeat this process across several folds and select the $K$ that yields the highest average validation score. This provides a simple data-driven way to select $K$ without using labeled target data. Figure 7(b) shows the validation accuracy obtained from 5-fold cross validation. Consistent with the results in Figure 7(a), performance improves as $K$ increases, and a stable region emerges when $K \ge 9$ with $\epsilon_2$ in the range $0.4$-$0.8$. These patterns suggest that choosing $K$ within the range of $9$ to $14$ is a reasonable option. Moreover, the maximin effect empirically recommends using a moderate number of pseudo-groups, typically between $5$ and $20$, which aligns well with our observations.
>
>
> - **Sensitivity analysis**:  Finally, to assess the robustness of our proposed method and verify the stability of $K$, we present a sensitivity analysis on the test set. Figure 7(c) summarizes the test accuracy across different values of $K$ and $\epsilon_2$. Across all choices of $\epsilon_2$, performance varies smoothly, indicating that the method is not highly sensitive to this parameter. As $K$ increases, accuracy improves steadily up to $K=7$, after which the results plateau and remain within a broad high-performing range. A stable region appears for $K \ge 8$, where all settings yield consistently strong performance between 84\% and 86\%. These patterns indicate that the method does not rely on a narrowly tuned value of $K$ and that a moderate range of choices provides comparable results.
>
> We note that both our selection strategies lead to consistent choices of $K$ between $9$-$12$, further supporting the robustness of this procedure.
> While larger values of $K$ may offer marginal accuracy gains, we avoid choosing excessively large $K$ to balance performance with computational cost. We have incorporated this discussion into Appendix A.5 of the revised paper.

---

> ### Author Response · Authors · 2025-11-20
>
> **W2**
> > Discussion of the papers suggested by the reviewer
>
> We appreciate the reviewer for pointing out these recent papers. We are already aware of both [2] and [3]. Nevertheless, for the reasons detailed below, these approaches are not directly comparable to our setting.
>
> First, both methods are tailored exclusively for multi-source domain adaptation and rely on structural heterogeneity across multiple source domains. Our formulation can be extended to multi-source settings, but this paper focuses on the single-source scenario to avoid unnecessary complexity for the reader. In the single-source setting, such cross-domain heterogeneity does not exist, and therefore these methods cannot serve as meaningful baselines for our experiments.
>
> Second, these approaches fall under the broader line of causal representation learning, originating from the framework of “Variational Autoencoders and Nonlinear ICA.” This line of work assumes an anticausal generative model in which latent variables—some invariant and others environment-specific—generate the observed features. Identifying the invariant components requires explicit environment indices and a sufficiently large number of heterogeneous environments. Moreover, these methods are generative: they aim to recover the full data distribution and therefore require learning both an encoder and a decoder. As a result, they typically demand a large amount of unlabeled target data to estimate the target marginal distribution and identify latent factors reliably.
>
> In contrast, our setting considers extremely limited unlabeled target samples and does not assume access to environment labels. Under such target-data scarcity, the key assumptions underlying these causal representation learning approaches break down. For example, [2] relies on an accurate estimate of the target marginal $P^{\mathrm{tg}}_X$ for partial identifiability, while [3] requires stable estimation of domain-specific subspaces across multiple heterogeneous sources. Both requirements become infeasible in our regime, and their theoretical guarantees no longer hold.
>
> We thank the reviewer again for raising this connection. We added them to the related work section of the revised paper.
>
> ---
>
> **W3**
> > Concern about the relaxation gap in the surrogate objective
>
> Thank you for pointing out this important issue. The surrogate objective can indeed be strictly smaller than the original objective. As described in Section A.1, the gap arises from the inequality:
> $$
> \sup_{x: ||z(x) - z(X)||_2 \leq \epsilon_1} \mathbb{E} _{Q _{Y|X}^\beta}\big[ \ell (f^{\theta}_Z(z(x)), Y) \big] \leq \sup _{z' : ||z' - z(X)||_2  \leq \epsilon_1}  \mathbb{E} _{Q _{Y|X}^\beta}  \big[ \ell (f^{\theta}_Z(z'), Y) \big].
> $$
>
> While we agree that the inequality may be strict in general when the feature map $z(\cdot)$ is not surjective onto the $\epsilon_1$-ball, we believe that this phenomenon is negligible in practice for embedding distributions arising in modern deep networks. This is because it is widely assumed that the pushforward distribution of $X$ under the embedding $z(\cdot)$ possesses a positive Lebesgue density on, or in a neighborhood of, its effective support.
>
> Although raw images often concentrate near a low-dimensional manifold in the input space and therefore may not admit a Lebesgue density, the situation is different after they pass through several nonlinear layers. Empirically and conceptually, internal representations produced by modern deep networks behave as “thickened” manifolds and are typically modeled as distributions with nonzero volume in the ambient space. A notable example supporting this view is the widespread use of the Fréchet Inception Distance (FID) for evaluating generative models, which implicitly treats the distribution of the embedding as having a non-degenerate density in feature space.
>
> Formally proving the existence of a density for arbitrary deep embeddings is challenging. However, once we adopt this commonly held assumption, as is standard in practice, the relaxation introduced after Lemma A.1.1 becomes nearly tight. In such cases, the overlap between the $\epsilon_1$-ball and the image of $z(\cdot)$ is substantial, making the resulting upper bound a close approximation of the original hierarchical objective.
>
> We added a brief clarification in Appendix A.1.1 in the revised version to make this point explicit.
>
> ---
> **Question**
> >Clarification of lines 1004-1007
>
> This was a typo, and the correct term in lines 1004–1005 should be $\epsilon_1$, not $\epsilon_2$. We corrected this in the revision.
>
> ---
>
> **Reference**
>
> [1] Meinshausen and Bühlmann. Maximin effects in inhomogeneous large-scale data. AOS, 2015.
>
> [2] Kong et al. Partial identifiability for domain adaptation. ArXiv:2306.06510, 2023.
>
> [3] Li et al. Subspace identification for multi-source domain adaptation. NeurIPS, 2023.
>
> ---
>
> We sincerely thank the reviewer once again and hope that our responses adequately address the concerns.

---

### Official Review · Reviewer_Lo6k · 2025-10-29

**Soundness:** 2
**Presentation:** 3
**Contribution:** 2
**Rating:** 4
**Confidence:** 4

**Summary:**

The paper proposes a distributionally robust optimization (DRO) framework for unsupervised domain adaptation (UDA), particularly under multi-source and target-data-scarce settings. The method defines an ambiguity set over both the covariate (input) distribution and the conditional label distribution, allowing robustness against (i) uncertainty in target inputs and (ii) uncertainty in which source conditional distributions to rely on. The framework is applicable to both multi-source and single-source UDA, using pseudo-sources generated via sub-sampling. A tractable minimax algorithm is derived (Eqs. (3)–(7); Algorithm 1), optimizing over feature perturbations, mixture weights, and classifier parameters. Experiments on digit datasets (MNIST, SVHN, USPS) and spurious-correlation benchmarks (Waterbirds, CelebA, CMNIST) demonstrate consistent performance gains over UDA and robust-learning baselines (Tables 1–2).

**Strengths:**

1.	The paper identifies two overlooked issues in UDA: scarce unlabeled target data and spurious source correlations.
2.	This dual modeling is theoretically elegant and practically relevant for multi-source robustness.
3.	The paper is well-written and easy to read.

**Weaknesses:**

1.	D₁ (Wasserstein-∞) and D₂ (Euclidean) are chosen “for computational tractability” (Sec. 3.3) but without theoretical or empirical justification.
2.	Hyperparameters ϵ₁, ϵ₂ are selected via a small labeled validation set (Sec. 4.1), partially violating the unsupervised setting.
3.	No ablation compares using only conditional-mixing vs. only covariate-perturbation.
4.	The relationship between pseudo-source construction (Sec. 3.1) and mixture weights β is not fully explained; readers may confuse stochastic sub-sampling with real domain partitioning.
5.	Absence of gradient-stability discussion under joint optimization may raise reproducibility concerns.
6.	Compared to DRO literature [a, b], the paper’s theoretical contribution is weak.

[a] Learning models with uniform performance via DRO.
[b] Distributionally robust stochastic optimization with Wasserstein distance.

**Questions:**

Please see the weaknesses.

---

> ### Author Response · Authors · 2025-11-20
>
> We thank the reviewer for the careful reading of our paper. Below we provide a point-by-point response, and all corresponding revisions have been incorporated into the revised paper and highlighted in blue.
>
> ---
> **W1**
> > Question about the choice of $D_1$ and $D_2$
>
> We thank the reviewer for raising this important point. The choice of ambiguity sets is indeed a central design decision in DRO, and various constructions are possible depending on the modeling goal and computational considerations. Below, we summarize the specific ambiguity-set choices made in our framework and the rationale behind each of them. We hope this provides a clear explanation of our design choices.
>
> **(1) Pixel space vs. feature space**
>
> Although this point was not explicitly raised by the reviewer, we would like to emphasize that a key design decision in DRO is whether the ambiguity set is defined in the pixel/data space or in the feature/embedding space. In vision tasks, meaningful target-domain shifts typically correspond to geometric or semantic variations (e.g., digit-shape changes or background and pose variations in Waterbirds). Prior work (e.g., [1]) has empirically demonstrated that modeling distributional uncertainty in the feature space is more effective than in the pixel space for capturing such high-level shifts. For this reason, our method constructs the ambiguity set in the feature space.
>
> **(2) Wasserstein vs. $f$-divergence**
>
> While the Wasserstein distance and stronger (pseudo-)metrics such as the total variance and $f$-divergence are topologically very different, prior work has shown that, when restricted to sufficiently smooth densities, these discrepancies become much less pronounced [2,3]. Since our method operates in the feature space—where deep representations are often assumed to induce smoother and more semantically organized distributions—we expect that, at the population level, the difference between Wasserstein- and $f$-divergence–based ambiguity sets would not be substantial.
>
> In practice, however, the population distribution is unknown, and ambiguity sets must be constructed from the empirical measure. A well-known limitation of $f$-divergence–based ambiguity sets is that they require the target distribution to share support with the empirical distribution. This constraint restricts the ambiguity set to reweightings of the empirical measure. When the number of target samples is sufficiently large, this restriction is mild, and in such settings we indeed expect Wasserstein and $f$-divergence DRO formulations to behave similarly.
>
> However, when the target sample size is small, this constraint becomes problematic: it prevents the ambiguity set from including plausible target distributions whose support deviates from the observed empirical samples. This phenomenon has been widely discussed in recent DRO literature and is one of the main reasons why Wasserstein-based ambiguity sets are generally preferred in modern machine-learning applications (e.g., [4,5,6]).
>
> **(3) Finite-order vs. infinite-order**
>
> Although $W_\infty$ is topologically strictly stronger than $W_p$ for any finite $p$, our experience—as well as prior empirical observations—suggests that, when applying DRO in deep learning, using $W_p$ or $W_\infty$ often leads to similar robustness behavior in practice. On the other hand, the computational difference between these two cases is substantial. Because $W_\infty$ leads to a much simpler and more stable optimization procedure, we adopt $W_\infty$ in our framework.
>
> More precisely, the primal Wasserstein DRO objective (P) admits the following standard dual formulation (D) [7]:
> $$
> (P) : \sup_{Q: W_p(P,Q) \le \epsilon} \mathbb{E}_{(X,Y)\sim Q} \left[\ell(f^\theta(X), Y)\right]
> $$
>
> For $p\in[1,\infty)$,
> $$(D): \min_{\lambda\ge 0} \Big[ \lambda \rho^p + \mathbb{E}_{(X,Y) \sim P} \Big[ \sup _{x' \in \mathcal{X}}  \\{\ell(f^\theta(x'),Y) - \lambda\, c(X,x')^p \\} \Big]$$
>
> For $p=\infty$,
> $$(D): \mathbb{E}_{(X,Y)\sim P}\Big[\sup _{x': c(X,x') \le \epsilon}\ell(f^\theta(x'), Y) \Big],$$
>
> where $c(\cdot)$ denotes the cost function. This formulation clearly shows the structural gap between finite-order and infinite-order cases. For $p<\infty$, the dual contains an inner maximization over $x'$ and an outer minimization over $\lambda$, resulting in an iterative bi-level optimization at every training step. This significantly increases computational overhead and often leads to instability in deep neural networks.
>
> In contrast, when $p=\infty$, the dual simplifies to $\sup_{x': c(X,x')\le\epsilon} \ell(f^\theta(x'),Y)$ which can be solved directly with a single projected-gradient ascent step of radius $\epsilon$. The update formulas in Eq. (3) and Eq. (5) in Section 3.4 leverage exactly this property, enabling efficient and stable training in high-dimensional settings. This computational advantage is one of the main reasons why many practical DRO methods in deep learning adopt $W_\infty$ [8,9].

---

> ### Author Response · Authors · 2025-11-20
>
> (continued)
>
> **(4) The choice of $L_2$ for $D_2$**
>
>  In the same manner, we use the $L_2$ distance for $D_2$ due to its computational simplicity.
> The $L_2$ ball admits an analytic projection, which leads to efficient and straightforward
> updates of mixture weights as in Eq. (6). This choice is consistent with prior work that
> adopts Euclidean constraints on the simplex [10].
>
> ---
>
> **W2**
> > Concern about the validation via a small labeled target data
>
> Thank you for raising this important point. In UDA, selecting hyperparameters is fundamentally ill-posed because no labeled target data are available; without such supervision, there is no widely accepted or theoretically grounded way to choose hyperparameters that influence target-domain performance. Although several heuristic strategies exist—such as source-only cross-validation with partitions designed through simple heuristic rules, reverse validation, or confidence-based scoring—these methods often require enough target data to work reliably and can become unstable when the target data are very limited.
>
> To ensure a fair and consistent comparison with prior UDA methods, we followed the widely adopted evaluation protocol that assumes access to a small labeled target validation set. This practice is standard in the UDA literature; representative methods such as DANN [11], IRM [12], MCD [13], ICON [14], and many others rely on the same setting because a reliable unsupervised model-selection criterion is currently unavailable. We therefore adopted this well-established protocol to avoid confounding effects from unstable heuristics and to enable fair comparisons across methods.
>
> We nevertheless acknowledge the reviewer’s concern and therefore further evaluate our method using the well-known *leave-one-domain-out* cross-validation [15]. This procedure operates as follows: given $K$ source domains, we train $K$ models using the same hyperparameters, each time holding out one source domain as validation. Each model is then evaluated on its held-out domain, and we compute the average accuracy over all held-out domains. The hyperparameters maximizing this average accuracy are finally selected, and a new model is retrained on all $K$ domains.
>
> Table 1 and Table 2 below report the test accuracies of our method tuned with a small labeled target validation set, denoted by Ours($\cdot$), and those tuned via leave-one-domain-out cross validation, denoted by *Ours($\cdot$). Although *Ours($\cdot$) shows a moderate drop in accuracy compared to Ours($\cdot$), the results remain consistently reasonable across all benchmark settings and still outperform the other baselines. These results demonstrate that our method does not critically depend on labeled target validation data and remains robust under a fully unsupervised hyperparameter selection strategy.
>
> We incorporate these additional results into the revised version of the paper, and we hope that this clarification adequately addresses the reviewer’s concern.

---

> > ### Author Response · Authors · 2025-11-20
> >
> > **Reference**
> >
> > [1] Volpi et al. Generalizing to unseen domains via adversarial data augmentation. NeurIPS, 2018.
> >
> > [2] Chae and Walker. Wasserstein upper bounds of the total variation for smooth densities. Statistics \& Probability Letters, 2020.
> >
> > [3] Chae. Wasserstein upper bounds of lp-norms for multivariate densities in besov spaces. Statistics \& Probability Letters, 2024.
> >
> > [4] Kuhn et al. Wasserstein distributionally robust optimization: Theory and applications in machine learning. Operations Research \& Management Science in the Age of Analytics, 2019.
> >
> > [5] Blanchet et al. Statistical analysis of Wasserstein distributionally robust estimators. Tutorials in Operations Research, 2021.
> >
> > [6] Gao and Kleywegt. Distributionally robust stochastic optimization with Wasserstein distance. Mathematics of Operations Research, 2023.
> >
> > [7] Gao et al. Wasserstein distributionally robust optimization and variation regularization. OR, 2024.
> >
> > [8] Staib and Jegelka. Distributionally robust deep learning as a generalization of adversarial training. NeurIPS Workshop on Machine Learning and Computer Security, 2017.
> >
> > [9] Balunovic et al. Certifying geometric robustness of neural networks. NeurIPS, 2019.
> >
> > [10] Wang et al. Distributionally robust machine learning with multi-source data. AOS, 2025.
> >
> > [11] Ganin et al. Domain-adversarial training of neural networks. JMLR, 2016.
> >
> > [12] Arjovsky et al. Invariant risk minimization. ArXiv:1907.02893, 2019.
> >
> > [13] Saito et al. Maximum classifier discrepancy for unsupervised domain adaptation. CVPR, 2018.
> >
> > [14] Yue et al. Make the U in UDA matter: Invariant consistency learning for unsupervised domain adaptation. NeurIPS, 2023.
> >
> > [15] Gulrajani and Lopez-Paz. In search of lost domain generalization. ArXiv:2007.01434, 2020.
> >
> > [16] Meinshausen and Bühlmann. Maximin effects in inhomogeneous large-scale data. AOS, 2015.
> >
> > [17] Sagawa et al. Distributionally robust neural networks for group shifts: On the importance of regularization for worst-case generalization. ArXiv:1911.08731, 2019.
> >
> > [18] Krueger et al. Out-of-distribution generalization via risk extrapolation (rex). ICLR, 2021.
> >
> > [19] Zhou et al. Distributionally robust multilingual machine translation. ArXiv:2109.04020, 2021.
> >
> > [20] Jeong et al. Multi-expert distributionally robust optimization for out-of-distribution generalization. NeurIPS, 2025.

---

> ### Author Response · Authors · 2025-11-20
>
> **Table 1.** Comparison of test accuracies on digit benchmarks across different target sample sizes.
>
> ---
>
> | Method        |SVHN→MNIST |         | MNIST→USPS |           | USPS→MNIST |       |
> |---------------|----------|-------|----------|-------|----------|-------|
> |               |    10²      |   10    | 10²         | 10    | 10²         | 10    |
> | ERM (Src)     | 59.6 ± 1.8  | 59.6 ± 1.8 | 63.4 ± 1.4 | 63.4 ± 1.4 | 60.4 ± 5.7 | 60.4 ± 5.7 |
> | DANN          | 66.0 ± 4.9  | 61.2 ± 1.8 | 82.0 ± 3.9 | 74.3 ± 5.7 | 74.8 ± 6.7 | 51.1 ± 5.0 |
> | CDAN          | 63.4 ± 1.8  | 56.9 ± 0.9 | 80.8 ± 1.4 | 62.0 ± 1.8 | 58.3 ± 4.6 | 54.8 ± 5.9 |
> | MK-MMD        | 50.0 ± 3.0  | 48.3 ± 1.0 | 63.3 ± 1.1 | 41.3 ± 5.5 | 57.6 ± 3.2 | 32.1 ± 3.9 |
> | ATDOC         | 83.3 ± 9.1  | 59.1 ± 5.7 | 91.1 ± 0.8 | 73.6 ± 4.2 | 92.6 ± 1.3 | 87.0 ± 3.6 |
> | STAR          | 76.4 ± 1.5  | 66.8 ± 0.9 | 90.3 ± 1.9 | 81.3 ± 7.6 | 94.5 ± 0.7 | 85.2 ± 2.7 |
> | CORAL         | 75.4 ± 2.7  | 63.6 ± 0.9 | 90.4 ± 0.7 | 85.4 ± 0.5 | 75.9 ± 1.9 | 64.7 ± 3.5 |
> | MCD           | 79.1 ± 1.0  | 61.3 ± 0.7 | 89.3 ± 1.6 | 84.5 ± 2.3 | 96.1 ± 1.6 | 85.9 ± 4.0 |
> | Ours (ERM)    | 92.0 ± 1.6  | 87.0 ± 0.9 | 92.1 ± 1.1 | 87.1 ± 3.6 | 90.3 ± 2.2 | 86.4 ± 2.5 |
> | Ours (CDAN)   | 92.5 ± 2.3  | 87.0 ± 2.0 | 93.5 ± 1.3 | 87.8 ± 1.5 | 91.5 ± 1.9 | 87.0 ± 1.9 |
> | Ours (STAR)   | **94.4 ± 1.7** | **91.3 ± 1.1** | **95.6 ± 1.0** | **91.2 ± 1.4** | **97.3 ± 0.8** | **93.0 ± 2.8** |
> | *Ours (STAR) | 92.9 ± 2.1  | 85.1 ± 1.2 | 93.6 ± 1.5 | 90.5 ± 0.6 | 96.6 ± 1.4 | 87.6 ± 3.2 |
>
> ---
>
> **Table 2.** Comparison of test accuracies across spurious benchmarks.
>
> ---
>
> | Method                | Group |      | Waterbirds        |      | CelebA            |      | CMNIST          |      |
> |-----------------------|--------|------|--------------------|------|--------------------|------|------------------|------|
> |                       | label |      | Test Acc           |      | Test Acc           |      | Test Acc         |      |
> | ERM (Src)             | ×      |      | 48.4 ± 0.9         |      | 35.5 ± 0.6         |      | 0.9 ± 0.5        |      |
> | DANN                  | ×      |      | 35.8 ± 4.5         |      | 23.5 ± 2.1         |      | 0.9 ± 1.8        |      |
> | CDAN                  | ×      |      | 46.2 ± 1.8         |      | 24.6 ± 1.5         |      | 1.2 ± 0.4        |      |
> | MK-MMD                | ×      |      | 45.1 ± 1.2         |      | 27.7 ± 2.9         |      | 2.8 ± 1.3        |      |
> | ATDOC                 | ×      |      | 47.3 ± 1.4         |      | 31.8 ± 1.4         |      | 3.1 ± 0.9        |      |
> | STAR                  | ×      |      | 49.8 ± 5.6         |      | 24.4 ± 2.4         |      | 2.2 ± 2.7        |      |
> | CORAL                 | ×      |      | 50.9 ± 2.9         |      | 31.7 ± 1.9         |      | 1.7 ± 0.4        |      |
> | MCD                   | ×      |      | 59.0 ± 3.1         |      | 30.7 ± 2.5         |      | 1.9 ± 1.9        |      |
> | DRST                  | ×      |      | 37.1 ± 6.0         |      | 29.5 ± 4.6         |      | 1.0 ± 0.7        |      |
> | ICON                  | ×      |      | 54.2 ± 1.4         |      | 31.1 ± 2.7         |      | 4.4 ± 3.2        |      |
> | GroupDRO              | ✓      |      | 61.4 ± 2.7         |      | 63.0 ± 2.6         |      | 3.4 ± 1.6        |      |
> | GroupDRO (Tgt)        | ✓      |      | 90.6 ± 0.2         |      | 89.3 ± 1.3         |      | 73.1 ± 0.3       |      |
> | PDE                   | ✓      |      | 57.1 ± 6.6         |      | 55.0 ± 5.5         |      | 1.3 ± 1.2        |      |
> | Ours (ERM)            | ×      |      | **87.3 ± 2.1**     |      | **85.0 ± 4.1**     |      | **7.5 ± 0.5**    |      |
> | *Ours (ERM)          | ×      |      | 83.3 ± 2.7         |      | 76.0 ± 3.8         |      | 4.9 ± 0.7        |      |

---

> ### Author Response · Authors · 2025-11-20
>
> **W3**
> >No ablation compares using only conditional-mixing vs. only covariate-perturbation.
>
> We thank the reviewer for the insightful question. Although the main paper does not include a standalone ablation, the individual effects of covariate perturbation and conditional mixing can be directly examined through the heatmaps in Figure 1(https://drive.google.com/file/d/1DaDQtCN-q3XSPpiQk-NHRUypXWFeVi_1/view?usp=drive_link) presents the heatmaps of average test accuracy across $(\epsilon_1, \epsilon_2)$.
> Figure 1(a) corresponds to the MNIST$\rightarrow$USPS task with $10^2$ unlabeled target samples,
> and Figure 1(b) corresponds to the USPS$\rightarrow$MNIST task with $10$ unlabeled target samples.
> The following empirical observations clarify the detailed effects of ($\epsilon_1$, $\epsilon_2$).
>
> - Both $\epsilon_1$ and $\epsilon_2$ yield substantial improvement over the baseline. The baseline setting $(\epsilon_1,\epsilon_2)=(0,0)$ performs noticeably worse than configurations that introduce moderate uncertainty. Increasing either $\epsilon_1$ or $\epsilon_2$ generally improves accuracy, indicating that controlled covariate or conditional perturbations enhance generalization. These patterns show that the two components of the ambiguity set contribute complementary robustness, with performance gains observed across a broad region of hyperparameter values.
>
> - When the target sample size is moderately large, both hyperparameters exhibit a broad region of stability. In Figure 1(a), target accuracy remains high across a wide plateau of $(\epsilon_1,\epsilon_2)$ values. For example, $\epsilon_1 \in \\{0.2, 0.4\\}$ combined with $\epsilon_2 \ge 0.2$ consistently results in strong performance. This plateau suggests insensitivity to small variations in the hyperparameters and indicates that the method does not rely on precise tuning.
>
> - Under extreme target-data scarcity, the effect of $\epsilon_1$ naturally becomes dominant.  As shown in Figure 1(b), target accuracy changes sharply along the $\epsilon_1$ axis (particularly for $\epsilon_1 \ge 0.6$), while remaining relatively flat along $\epsilon_2$. This phenomenon is consistent with the problem's nature: when $\hat{P}^{\textrm{tg}}_X$ is estimated from very few samples, covariate-level variability dominates, and robustness to input perturbations becomes critical.
> \eed
>
> In addition, we emphasize that the appropriate value of $\epsilon_1$ depends on the scale of the learned embedding, and thus no universal choice applies across models.
> What matters in practice is that accuracy remains stable over a wide range of $\epsilon_1$, as observed in Figure 1. This stability suggests that heuristic choices for $\epsilon_1$ work well in practice. In contrast, $\epsilon_2$ can be set to a relatively large value (e.g., $\infty$) when no prior knowledge is available, as conditional mixing does not induce instability.
>
> We have incorporated these clarifications and the accompanying empirical results into Section 4.4 of the revised paper.
>
> ---
>
> **W4**
> >Clarifying the relationship between pseudo-source construction and mixture weights $\beta$
>
> We agree that the connection between pseudo-source construction and the mixture weights $\beta$ may be confusing in the current version, and we revised the paper to clarify this relationship more explicitly.
>
> We first clarify that the pseudo-source construction in Section 3.1 is not intended to represent real domain partitioning. In the single-source setting, random subsampling is used to extract diverse latent conditional structures that are contained in the source distribution. This follows the idea of the maximin effect [16], where different subsamples provide different conditional estimators. In a multi-source setting, the conditionals estimated from each explicit (real) source domain can be used directly.
>
> The role of $\beta$ is to form an adversarial mixture over these estimated conditionals. Specifically, the update rule in Eq. (6) assigns larger weights to conditional estimators that induce higher loss under the current classifier $f^{\theta}$. The model parameters $\theta$ are then updated to minimize the resulting adversarial objective. By iterating these updates, the classifier becomes robust to conditional uncertainty and potential mixture shifts.

---

> ### Author Response · Authors · 2025-11-20
>
> **W5**
> >Absence of gradient-stability discussion under joint optimization
>
> We thank the reviewer for raising this important concern. To examine the gradient stability of the joint optimization procedure, we performed an additional controlled experiment. In this experiment, we empirically analyzed the gradient trajectories of both the $\beta$-gradient (adversarial weights) and the $\theta$-gradient (model parameters).
>
> Specifically, we fixed the hyperparameters $(\epsilon_1, \epsilon_2) = (0.6, 0.4)$ and performed a domain adaptation experiment on the SVHN$\rightarrow$MNIST task using a mini-batch size of 128. At each iteration, we computed the $L_2$ norm of the gradients with respect to $\beta$ and $\theta$. For panel (a), the learning rate for $\theta$ was fixed at $10^{-3}$ while varying the learning rate for $\beta$. For panel (b), the learning rate for $\beta$ was fixed at $10^{-2}$ while varying the learning rate for $\theta$. For each learning rate in $\\{10^{-1}, 10^{-2}, 10^{-3}\\}$, we repeated the procedure 20 times to assess stability and convergence across different update magnitudes.
>
> Figure 6 (https://drive.google.com/file/d/13_dIgO51_LE_GvfACe8mLTena0o0TYlV/view) in the revised main paper demonstrates the resulting gradient trajectories of $\beta$ and $\theta$, respectively.
> Several observations are noteworthy.
>
> - Across all learning rates, both $\beta$ and $\theta$ show monotonically decreasing and smooth gradient norms, indicating that the updates consistently move toward a stable point without sharp fluctuations.
>
> - Even under the most aggressive learning rate ($10^{-1}$), the gradient norms remain well-controlled and do not exhibit any signs of gradient explosion, oscillation, or unstable plateauing, which are typical indicators of optimization instability in min–max procedures.
>
> - The variance bands across 20 random repetitions are uniformly narrow, suggesting that the optimization dynamics are highly reproducible and not sensitive to stochastic variation in initialization or minibatch sampling.
>
> These empirical findings demonstrate that the proposed joint optimization procedure is numerically stable, robust to learning-rate variation, and reliable across repeated runs.
>
> We further emphasize that our learning algorithm is an extension of the standard GroupDRO framework [17]. The convergence behavior of GroupDRO has been extensively analyzed, and its practical stability is well established in the literature. In addition, many subsequent works have adopted similar min–max optimization structures and have consistently reported stable and reliable training dynamics across diverse architectures and datasets [18,19,20].
>
> We have incorporated these clarifications and the accompanying empirical results into Appendix A.4 of the revised paper.
>
> ---
>
> **W6**
> > Weak theoretical contribution
>
> We appreciate the reviewer’s observation. The two referenced papers indeed provide strong theoretical foundations, and we agree that further theoretical analysis of our framework would be valuable. At the same time, we believe that a full theoretical development is beyond the scope of this short conference paper, whose primary goal is to introduce a practical and effective methodology. We hope that the extensive empirical results across multiple real-world datasets provide convincing evidence of the effectiveness of our approach, and we view a deeper theoretical treatment as an important direction for future work.
>
> ---
>
> We sincerely thank the reviewer once again for the valuable comments, and we hope that our responses adequately address the concerns.

---

### Official Review · Reviewer_Fp6T · 2025-10-29

**Soundness:** 3
**Presentation:** 3
**Contribution:** 3
**Rating:** 4
**Confidence:** 4

**Summary:**

This paper proposes a novel distributionally robust optimization (DRO) framework for multi-source unsupervised domain adaptation (MS-UDA). The method models both input marginal distribution shifts and label conditional distribution shifts by constructing a mixture-based ambiguity set that combines conditional distributions from multiple source domains with adaptive weighting, while allowing controlled perturbations of the target input distribution. The authors further design a minimax optimization algorithm that alternately updates feature perturbations, mixture weights, and classifier parameters. Experimentally, results on several benchmark datasets (MNIST/SVHN/USPS, Waterbirds, CelebA, CMNIST) demonstrate that this approach achieves significantly better classification performance than mainstream UDA methods such as DANN, CDAN, and STAR, particularly in scenarios with scarce target data or spurious correlations.

**Strengths:**

1. The paper proposes a unified Distributionally Robust Optimization (DRO) framework that is applicable to both multi-source and single-source Unsupervised Domain Adaptation (UDA), offering flexibility across different scenarios.

2. Unlike traditional approaches that typically model either input uncertainty or label distribution uncertainty in isolation, this method simultaneously accounts for both, providing a more comprehensive solution.

3. The proposed algorithm is highly tractable, ensuring that it can be seamlessly integrated with existing UDA frameworks, making it easy to adopt and implement in real-world applications.

4. The experimental results demonstrate the method's robust performance, especially in tasks where target data is scarce or where spurious correlations are prevalent, outperforming several strong baselines.

**Weaknesses:**

1. A reliance on labeled target data for model selection moves the problem into a "partially supervised" or "few-shot" adaptation setting. While the main training uses unlabeled target data, the crucial choice of hyperparameters $\epsilon_1$ and $\epsilon_2$ is supervised.

2. The appendix shows heatmaps of performance vs. $(\epsilon_1, \epsilon_2)$, which is good. However, the main paper should discuss how sensitive the method is. Is there a broad range of "good" hyperparameters, or does the performance collapse without precise tuning? This context is crucial.

3. Although multiple standard datasets are used for testing, there is a lack of evaluation in broader domains (such as NLP or time-series data) to demonstrate the generalizability of the method.

4. The conclusion section does not sufficiently discuss the limitations of the method and potential future directions for improvement. Adding these aspects would help present a more comprehensive view of the research depth.

**Questions:**

1. How does the reliance on labeled target data for hyperparameter selection (specifically for $\epsilon_1$ and $\epsilon_2$) affect the generalization of the method in unsupervised domain adaptation? Could this be considered as a shift towards a "partially supervised" or "few-shot" adaptation setting?

2. How sensitive is the proposed method to these hyperparameters?

3. How does the proposed method perform on data from other modalities?

4. How critical is the choice of base classifier (ERM vs. CDAN/STAR) for the final performance?

---

> ### Author Response · Authors · 2025-11-20
>
> We thank the reviewer for the careful reading of our paper. Below we provide a point-by-point response, and all corresponding revisions have been incorporated into the revised paper and highlighted in blue.
>
> ---
>
> **W1, Q1.**
> > Concern about the reliance on labeled target data for validation
>
> Thank you for raising this important point. In UDA, selecting hyperparameters is fundamentally ill-posed because no labeled target data are available; without such supervision, there is no widely accepted or theoretically grounded way to choose hyperparameters that influence target-domain performance. Although several heuristic strategies exist—such as source-only cross-validation with partitions designed through simple heuristic rules, reverse validation, or confidence-based scoring—these methods often require enough target data to work reliably and can become unstable when the target data are very limited.
>
> To ensure a fair and consistent comparison with prior UDA methods, we followed the widely adopted evaluation protocol that assumes access to a small labeled target validation set. This practice is standard in the UDA literature; representative methods such as DANN [1], IRM [2], MCD [3], ICON [4], and many others rely on the same setting because a reliable unsupervised model-selection criterion is currently unavailable. We therefore adopted this well-established protocol to avoid confounding effects from unstable heuristics and to enable fair comparisons across methods.
>
> We nevertheless acknowledge the reviewer’s concern and therefore further evaluate our method using the well-known *leave-one-domain-out* cross validation protocol [5]. This procedure operates as follows: given $K$ source domains, we train $K$ models using the same hyperparameters, each time holding out one source domain as validation. Each model is then evaluated on its held-out domain, and we compute the average accuracy over all held-out domains. The hyperparameters maximizing this average accuracy are finally selected, and a new model is retrained on all $K$ domains.
>
> Table 1 and Table 2 (same numbering as in the revised paper) below report the test accuracies of our method tuned with a small labeled target validation set, denoted by Ours($\cdot$), and those tuned via leave-one-domain-out cross validation, denoted by *Ours($\cdot$). Although *Ours($\cdot$) exhibits a moderate performance drop compared to Ours($\cdot$), the results remain consistently competitive across all benchmark settings. Also, *Ours($\cdot$) still outperforms them even though the competing baselines are tuned using labeled target data. These results demonstrate that our method remains robust under a fully unsupervised hyperparameter selection strategy.
>
> We incorporated these additional results into Section 4 of the revised version of the paper, and we hope that this clarification adequately addresses the reviewer’s concern.

---

> ### Author Response · Authors · 2025-11-20
>
> **Table 1.** Comparison of test accuracies on digit benchmarks across different target sample sizes.
>
> ---
>
> | Method        |SVHN→MNIST |         | MNIST→USPS |           | USPS→MNIST |       |
> |---------------|----------|-------|----------|-------|----------|-------|
> |               |    10²      |   10    | 10²         | 10    | 10²         | 10    |
> | ERM (Src)     | 59.6 ± 1.8  | 59.6 ± 1.8 | 63.4 ± 1.4 | 63.4 ± 1.4 | 60.4 ± 5.7 | 60.4 ± 5.7 |
> | DANN          | 66.0 ± 4.9  | 61.2 ± 1.8 | 82.0 ± 3.9 | 74.3 ± 5.7 | 74.8 ± 6.7 | 51.1 ± 5.0 |
> | CDAN          | 63.4 ± 1.8  | 56.9 ± 0.9 | 80.8 ± 1.4 | 62.0 ± 1.8 | 58.3 ± 4.6 | 54.8 ± 5.9 |
> | MK-MMD        | 50.0 ± 3.0  | 48.3 ± 1.0 | 63.3 ± 1.1 | 41.3 ± 5.5 | 57.6 ± 3.2 | 32.1 ± 3.9 |
> | ATDOC         | 83.3 ± 9.1  | 59.1 ± 5.7 | 91.1 ± 0.8 | 73.6 ± 4.2 | 92.6 ± 1.3 | 87.0 ± 3.6 |
> | STAR          | 76.4 ± 1.5  | 66.8 ± 0.9 | 90.3 ± 1.9 | 81.3 ± 7.6 | 94.5 ± 0.7 | 85.2 ± 2.7 |
> | CORAL         | 75.4 ± 2.7  | 63.6 ± 0.9 | 90.4 ± 0.7 | 85.4 ± 0.5 | 75.9 ± 1.9 | 64.7 ± 3.5 |
> | MCD           | 79.1 ± 1.0  | 61.3 ± 0.7 | 89.3 ± 1.6 | 84.5 ± 2.3 | 96.1 ± 1.6 | 85.9 ± 4.0 |
> | Ours (ERM)    | 92.0 ± 1.6  | 87.0 ± 0.9 | 92.1 ± 1.1 | 87.1 ± 3.6 | 90.3 ± 2.2 | 86.4 ± 2.5 |
> | Ours (CDAN)   | 92.5 ± 2.3  | 87.0 ± 2.0 | 93.5 ± 1.3 | 87.8 ± 1.5 | 91.5 ± 1.9 | 87.0 ± 1.9 |
> | Ours (STAR)   | **94.4 ± 1.7** | **91.3 ± 1.1** | **95.6 ± 1.0** | **91.2 ± 1.4** | **97.3 ± 0.8** | **93.0 ± 2.8** |
> | *Ours (STAR) | 92.9 ± 2.1  | 85.1 ± 1.2 | 93.6 ± 1.5 | 90.5 ± 0.6 | 96.6 ± 1.4 | 87.6 ± 3.2 |
>
> ---
>
> **Table 2.** Comparison of test accuracies across spurious benchmarks.
>
> ---
>
> | Method                | Group |      | Waterbirds        |      | CelebA            |      | CMNIST          |      |
> |-----------------------|--------|------|--------------------|------|--------------------|------|------------------|------|
> |                       | label |      | Test Acc           |      | Test Acc           |      | Test Acc         |      |
> | ERM (Src)             | ×      |      | 48.4 ± 0.9         |      | 35.5 ± 0.6         |      | 0.9 ± 0.5        |      |
> | DANN                  | ×      |      | 35.8 ± 4.5         |      | 23.5 ± 2.1         |      | 0.9 ± 1.8        |      |
> | CDAN                  | ×      |      | 46.2 ± 1.8         |      | 24.6 ± 1.5         |      | 1.2 ± 0.4        |      |
> | MK-MMD                | ×      |      | 45.1 ± 1.2         |      | 27.7 ± 2.9         |      | 2.8 ± 1.3        |      |
> | ATDOC                 | ×      |      | 47.3 ± 1.4         |      | 31.8 ± 1.4         |      | 3.1 ± 0.9        |      |
> | STAR                  | ×      |      | 49.8 ± 5.6         |      | 24.4 ± 2.4         |      | 2.2 ± 2.7        |      |
> | CORAL                 | ×      |      | 50.9 ± 2.9         |      | 31.7 ± 1.9         |      | 1.7 ± 0.4        |      |
> | MCD                   | ×      |      | 59.0 ± 3.1         |      | 30.7 ± 2.5         |      | 1.9 ± 1.9        |      |
> | DRST                  | ×      |      | 37.1 ± 6.0         |      | 29.5 ± 4.6         |      | 1.0 ± 0.7        |      |
> | ICON                  | ×      |      | 54.2 ± 1.4         |      | 31.1 ± 2.7         |      | 4.4 ± 3.2        |      |
> | GroupDRO              | ✓      |      | 61.4 ± 2.7         |      | 63.0 ± 2.6         |      | 3.4 ± 1.6        |      |
> | GroupDRO (Tgt)        | ✓      |      | 90.6 ± 0.2         |      | 89.3 ± 1.3         |      | 73.1 ± 0.3       |      |
> | PDE                   | ✓      |      | 57.1 ± 6.6         |      | 55.0 ± 5.5         |      | 1.3 ± 1.2        |      |
> | Ours (ERM)            | ×      |      | **87.3 ± 2.1**     |      | **85.0 ± 4.1**     |      | **7.5 ± 0.5**    |      |
> | *Ours (ERM)          | ×      |      | 83.3 ± 2.7         |      | 76.0 ± 3.8         |      | 4.9 ± 0.7        |      |

---

> ### Author Response · Authors · 2025-11-20
>
> **W2, Q2**
> >Sensitivity of the hyperparameters $(\epsilon_1, \epsilon_2)$
>
> In our framework, $\epsilon_1$ and $\epsilon_2$ control the levels of covariate uncertainty and conditional uncertainty, respectively. To directly address the reviewer's concern regarding sensitivity, we provide a refined analysis based on the heatmaps in Figure 1 (https://drive.google.com/file/d/1DaDQtCN-q3XSPpiQk-NHRUypXWFeVi_1/view?usp=drive_link) presents the heatmaps of average test accuracy across $(\epsilon_1, \epsilon_2)$.
> Figure 1(a) corresponds to the MNIST$\rightarrow$USPS task with $10^2$ unlabeled target samples,
> and Figure 1(b) corresponds to the USPS$\rightarrow$MNIST task with $10$ unlabeled target samples. The following observations clarify the role of these hyperparameters and show that the method does not depend on narrow tuning.
>
> -Both $\epsilon_1$ and $\epsilon_2$ yield substantial improvement over the baseline. The baseline setting $(\epsilon_1,\epsilon_2)=(0,0)$ performs noticeably worse than configurations that introduce moderate uncertainty. Increasing either $\epsilon_1$ or $\epsilon_2$ generally improves accuracy, indicating that controlled covariate or conditional perturbations enhance generalization. These patterns show that the two components of the ambiguity set contribute complementary robustness, with performance gains observed across a broad region of hyperparameter values.
>
> -When the target sample size is moderate, both parameters exhibit a wide stable region. In Figure 1(a), target accuracy remains high across a broad plateau of $(\epsilon_1,\epsilon_2)$ values. Specifically, the method exhibits a broad range of good hyperparameters: for example, $\epsilon_1 \in$ {0.2, 0.4} and $\epsilon_2 \ge 0.2$ both yield consistently high accuracy. This plateau indicates that the method is insensitive to small deviations from the chosen hyperparameters and does not hinge on precise tuning. This wide stable region supports the robustness of our method, showing that the model maintains strong performance under both types of uncertainty without exhibiting instability.
>
> -Under extreme target-data scarcity, the effect of $\epsilon_1$ naturally becomes dominant.
> As shown in Figure 1(b), target accuracy changes sharply along the $\epsilon_1$ axis (particularly for $\epsilon_1 \ge 0.6$), while remaining relatively flat along $\epsilon_2$. This phenomenon is consistent with the problem's nature: when $\hat{P}^{\textrm{tg}}_X$ is estimated from very few samples, covariate-level variability dominates, and robustness to input perturbations becomes critical.
>
> In addition, we emphasize that the appropriate value of $\epsilon_1$ depends on the scale of the learned embedding, and thus no universal choice applies across models.
> What matters in practice is that accuracy remains stable over a wide range of $\epsilon_1$, as observed in Figure 1. This stability suggests that heuristic choices for $\epsilon_1$ work well in practice. In contrast, $\epsilon_2$ can be set to a relatively large value (e.g., $\infty$) when no prior knowledge is available, as conditional mixing does not induce instability.
>
> We incorporated this clarification into Section 4.4 of the revised main paper, and we hope that the additional analysis fully resolves the concern.
>
> ---
>
> **W3, Q3**
> >Evaluation on broader domains.
>
> We appreciate the reviewer’s suggestion. Our current experimental setup follows standard practice in the UDA literature, focusing on widely used benchmarks where most prior work performs empirical comparisons and methodological analysis. We agree that evaluating broader domains is an important direction. However, extending our framework to NLP or time-series data requires domain-specific architectures and additional preprocessing steps. If we find this feasible within the rebuttal phase, we will attempt preliminary experiments and report the results.

---

> ### Author Response · Authors · 2025-11-20
>
> ---
>
> **W4**
> >Limited discussion of limitations and future directions.
>
> In the revised version, we will update the conclusion to briefly outline two natural future directions: (i) extending the framework to broader domains, and (ii) exploring alternative feature-space metrics or discrepancy measures beyond the current choices.
>
> ---
>
> **Q4**
> >Choice of base classifier (ERM vs. CDAN/STAR) for the final performance
>
> The choice of the base classifier is critical for the final performance of our method. As shown in Table 1 in the revised main paper, our method using stronger UDA classifiers such as CDAN or STAR leads to higher final performance compared to using ERM. In Table~2 in the revised main paper, our method with ERM outperforms our method combined with other UDA baselines because those UDA methods perform worse under spurious correlations. This reflects the fact that final performance largely depends on the capability of the underlying classifier. Our framework is classifier-agnostic and can be applied to any base classifier, making it readily compatible with state-of-the-art UDA methods. This flexibility allows the proposed DRO framework to further improve the performance of stronger classifiers without modifying their underlying architecture.
>
> ---
> We sincerely thank the reviewer once again for the valuable comments, and we hope that our responses adequately address the concerns.
>
> ---
>
> **Reference**
>
> [1] Ganin et al. Domain adversarial training of neural networks. JMLR, 2016.
>
> [2] Arjovsky et al. Invariant risk minimization. ArXiv:1907.02893, 2019.
>
> [3] Saito et al. Maximum classifier discrepancy for unsupervised domain adaptation. CVPR, 2018.
>
> [4] Yue et al. Make the U in UDA matter: Invariant consistency learning for unsupervised domain adaptation. NeurIPS, 2023.
>
> [5] Gulrajani and Lopez-Paz. In search of lost domain generalization. ArXiv:2007.01434, 2020.

---

### Official Review · Reviewer_nZt1 · 2025-11-03

**Soundness:** 3
**Presentation:** 2
**Contribution:** 3
**Rating:** 4
**Confidence:** 4

**Summary:**

The paper applies Distributionally Robust Optimization (DRO) to solve Unsupervised Domain Adaptation (UDA). It constructs ambiguity samples from the target data whose pseudolabel is the linear combination of the label from multiple source domain classifiers. The ambiguous sample is generated using a technique similar to adversarial training. The proposed method can also be viewed as a regularizer.

**Strengths:**

1. Novel extension of DRO to UDA
2. Impressive results on target accuracy

**Weaknesses:**

1. The paper attempts to apply Distributionally Robust Optimization (DRO) to the problem of Unsupervised Domain Adaptation (UDA). The idea of applying DRO to UDA appears to be novel although some of the technical contributions are from pre-existing DRO work. The proposed approach is not technically convincing due to tedious notation and workflow. It was quite difficult and frustrating to follow the logic in explanation in the paper.
2. Target accuracy is the only result made available. This does not give enough insight into the workings of the model.
3. What is the intuition behind Eq. 6. Whichever subset (domain) k has the least predictive power (largest loss) it gets the largest \beta_k. Why is this reasonable? Will this not result in negative transfer?
4. In Eq. 5, How is Euclidean projection onto set A equivalent to minimizing the Wasserstein distance D1. What do you mean by Euclidean projection? How is the \epsilon_1 constraint satisfied?

**Questions:**

In Eq 3. What is variable for the Expectation under \hat{P}^{tg}_X. Is it z’ or z(X)? Likewise, in the expectation In lines 775-777 what is the difference between z(x) and z(X). How does it change to z’ in lines 781-782. There is inadequate explanation.

---

> ### Author Response · Authors · 2025-11-20
>
> We thank the reviewer for the careful reading of our paper. Below we provide a point-by-point response, and all corresponding revisions have been incorporated into the revised paper and highlighted in blue.
>
> ---
>
> **W1.**
> > Weak presentation by notation and difficult workflow.
>
> We sincerely thank the reviewer for the feedback regarding the clarity of the presentation. We apologize that some parts of the notation or workflow were difficult to follow. While other reviewers (Lo6k and fB8c) found the paper clear and easy to read, we fully acknowledge that some explanations may still be improved, especially for readers less familiar with DRO-based UDA formulations.
>
> To address this, we will refine the exposition in the revision. If the reviewer could point to any specific places that were especially confusing, we would be very happy to revise those parts with priority. We appreciate the reviewer’s comments and will make the paper clearer accordingly.
>
>
> ---
>
> **W2.**
> > Target accuracy is the only result made available. This does not give enough insight into the workings of the model.
>
> We agree that target accuracy alone does not provide the full picture. At the same time, it remains the primary evaluation metric in the UDA literature, as it most directly reflects real-world performance under distribution shift. Representative works such as DAN [1], CDAN [2], MCD [3], ATDOC [4], and ICON [5] also rely on target accuracy as their main metric, and our evaluation follows this well-established convention.
>
> We also appreciate the reviewer’s suggestion to provide a broader view beyond reporting only the final test accuracy. To reflect this, we moved the ablation analysis from the Appendix into the main paper. Figure 1 (https://drive.google.com/file/d/1DaDQtCN-q3XSPpiQk-NHRUypXWFeVi_1/view?usp=drive_link) in revised main paper provides ablation analyses that visualize how the model behaves under different hyperparameter configurations. The heatmaps illustrate how varying $(\epsilon_1, \epsilon_2)$ influences robustness, and they clearly show how our ambiguity-set design improves upon the baseline $(\epsilon_1, \epsilon_2) = (0,0)$. This offers a more detailed view of the mechanism underlying our robustness gains. We expanded this discussion into Section 4.4 of the revised main paper to make the intuition and empirical behavior of the method clearer.
>
> ---
>
> **W3.**
> > Questions about the intuition behind Equation (6) and concerns regarding potential negative transfer.
>
> In our formulation, $\hat{p}^{(k)}_{Y\mid X}(\cdot \mid x)$ denotes the $k$-th source conditional distribution. In Eq. (6), the update does not increase the weight assigned to the source conditional with the least predictive power. Instead, it increases $\beta_k$ for the source conditional that induces the largest loss under the current model $f^\theta$. This corresponds to selecting the worst-case conditional distribution within the ambiguity set—the standard min–max structure in DRO for obtaining a robust estimator. In other words, the update identifies the most adversarial soft pseudo-label mixture at the current model $f^\theta$.
>
> As noted by the reviewer, allowing $\beta$ to vary freely over the full simplex could lead to an overly conservative estimator, especially in cases where the source and target distributions are very similar. For this reason, as shown in Eq. (2), we impose an $L_2$ constraint on $\beta$, which prevents collapse onto a single source conditional. The radius $\epsilon_2$ regulates the deviation from the reference vector $\bar{\beta}$ and mitigates the risk of negative transfer by preventing $\beta$ from concentrating excessively on a single source conditional distribution.
>
> ---
>
> **W4.**
> > Questions about the $D_1$ constraint.
>
> The connection between the Wasserstein $D_1$ constraint in Eq. (2) and the corresponding update rule in Eq. (5) follows directly from Proposition 3.1. Since we choose the cost function of the Wasserstein distance to be the Euclidean distance, the $D_1$ constraint can be rewritten as the Euclidean form shown in Eq. (3). This formulation naturally leads to the projected gradient–descent update in Eq. (5), which incorporates the $\epsilon_1$ constraint. We provide the complete derivation and explanation in the section “"Rational Behind Proposition 3.1” in Appendix A.1, where the update rule is shown to implement the constraint $D_1$ through our proposed mechanism.

---

> ### Author Response · Authors · 2025-11-20
>
> **Questions**
> > Clarification of several technical details
>
> In Eq. (3), the expectation is taken over the random variable $X$ under $\hat{P}^{\mathrm{tg}}_X$. The feature representation $z(X)$ is therefore the random quantity within the expectation and defines the center of the constraint set. In lines 775--777, the expression $\{x : ||z(x) - z(X) ||_2 \le \epsilon_1\}$ denotes all inputs $x$ whose feature representations $z(x)$ lie within an $L_2$-ball around the random center $z(X)$.
>
> In lines 781--782, instead of optimizing over such inputs $x$ in the data space, we consider feature values $z'$ directly for computational convenience. This substitution yields a tractable
> surrogate objective because optimizing over $z'$ under the same $L_2$ constraint avoids dealing with the potentially complex pre-image of $z(\cdot)$ in the input space.
>
> ---
>
> We sincerely thank the reviewer once again for the careful comments, and we hope that our responses adequately address the concerns.
>
> ---
> **Reference**
>
> [1] Long et al. Learning transferable features with deep adaptation networks. ICML, 2015.
>
> [2] Long et al. Conditional adversarial domain adaptation. NeurIPS, 2018.
>
> [3] Saito et al. Maximum classifier discrepancy for unsupervised domain adaptation. CVPR, 2018.
>
> [4] Tang et al. Unsupervised domain adaptation via structurally regularized deep clustering. CVPR, 2020.
>
> [5] Yue et al. Make the U in UDA matter: Invariant consistency learning for unsupervised domain adaptation. NeurIPS, 2023.

---

### Author Response · Authors · 2025-11-27

Dear Area Chair,

We sincerely appreciate your efforts in overseeing the review of our submission during this challenging review cycle. For your convenience, we provide below a concise summary of the reviewer discussion and the corresponding revisions made to the manuscript. All concerns raised by the reviewers, including those regarding methodology, notation, and empirical evaluation, have been carefully addressed in our rebuttal and incorporated into the revised paper.


---

**Summary of main concerns and responses**

---


> **(Reviewers nZt1, Lo6k)**: Clarity of the overall formulation, notation, and training workflow, including the roles of $D_1$, $D_2$, and the $\beta$ update.


- *Response*: We streamlined notation and added a clearer, step-by-step description of the ambiguity set, the constraints, and how the optimization proceeds from the original objective to the final algorithm (**Sections 3.1, 3.4; Appendix A.1**).


> **(Reviewers Fp6T, nZt1)**: Lack of a systematic robustness study for $\epsilon_1$ and $\epsilon_2$ and reliance on a single target accuracy metric.

- *Response*: We added detailed $(\epsilon_1,\epsilon_2)$ heatmaps and interpretations that show where performance is stable and how each radius affects robustness, providing a more informative view than a single accuracy number (**Section 4.4**).

> **(Reviewers Fp6T, Lo6k)**: Reliance on a small labeled target validation set and absence of a purely source-based model-selection strategy.

- *Response*: We introduced leave-one-domain-out cross validation using only source labels and reported the corresponding test results, showing that our method remains competitive without target labels for validation (**Section 4; Tables 1 and 2**).

> **(Reviewers Fp6T, fB8c)**: Insufficient discussion of broader applicability, limitations, base classifier choice, and relation to recent causal representation learning methods.

- *Response*: We clarified that the framework is classifier-agnostic, compared different base classifiers, and expanded the conclusion to discuss broader domains and limitations. We also explained that the suggested causal representation learning methods assume multiple labeled sources and abundant target data and are therefore not directly comparable to our single-source, scarce-target setting (**Sections 2, 4, 5**).

> **(Reviewer Lo6k)**: Unclear relationship between pseudo-source construction and $\beta$, and uncertainty about stability of the joint minimax optimization over $(\theta,\beta)$.

- *Response*: We clarified how pseudo-sources generate multiple conditional estimators and how Eq. (6) forms an adversarial mixture over them, and we added a gradient-norm study on SVHN$\to$MNIST showing stable optimization for both $\theta$ and $\beta$ (**Sections 3.1, 3.4; Appendix A.4**).

> **(Reviewer fB8c)**: Insufficient justification for the choice of the number of pseudo-groups $K$ and its sensitivity.

- *Response*: We described simple rules for choosing $K$ and added a sensitivity study over $(K,\epsilon_2)$, which shows that performance is stable once $K$ is moderately large (**Appendix A.5; Figure 7**).

> **(Reviewer fB8c)**: Need for clarification of the relaxation gap between the original and surrogate objectives.

- *Response:* We added a short explanation of the relaxation gap and clarified, using standard assumptions and empirical results, that this gap is small in our setting (**Appendix A.1.1**).

---

We hope that this summary clearly conveys how we have addressed the reviewers' concerns and strengthened both the clarity and empirical support of the paper, and we respectfully ask you to evaluate the revised manuscript in light of these revisions.

---

### Meta-Review · Area_Chair_4nyD · 2025-12-31

**Summary:**

This paper presented here is a distributionally robust optimization (DRO) framework for unsupervised domain adaptation (UDA), which pays special attention to the fact that both input distributions are uncertain and so is the conditional label distribution. The method combines covariate perturbations and integrates source domain distributions at each level with conditional mixing features into a tractable, simple minimax optimization algorithm. The paper offers empirical results to demonstrate significant performance improvement on various benchmark datasets, particularly in cases where there are few target data or there are spurious correlations.

**Reviewer Concerns:**

Reviewer nZt1 identified one major concern: lack of clarity in both notation and workflow. Reviewer Fp6T noted the absence of systematic robustness studies and a standard target accuracy metric. A number of reviewers also suggested that the use of a small labelled target validation set for the hyperparameter selection problem violates the unsupervised nature of the task. Some reviewers also find that this paper's conclusion is insufficient and lacks discussion of the method's limitations or whether it may have wider applications. A big concern of Reviewer fB8c regarding the choice of the number of pseudo-groups in a single-source scenario was not mentioned in the review. Moreover, Reviewer Lo6k expressed concerns about the stability of joint optimization. And several reviewers indicated the theoretical value of the paper was weaker compared to other DRO research.

**Reviewer Scores:**

Given the response, Reviewer nZt1 will most likely have raised the score. The addition of robustness analysis, leave-one-domain-out cross-validation, and detailed sensitivity analysis may cause Reviewer Fp6T to increase their score. Having elaborated on the limitations' discussion, the pseudo-source of your code, and the additional analysis of the stability of the gradient, the Reviewer Lo6k is also likely raise the score. This paper is now clearer and has addressed many of the previous concerns about methodology and experimentation. The authors effectively addressed the concerns about pseudo-groups, sensitivity, and gradient stability. The explanation about the surrogate gap, although not exhaustive, sufficiently reassured the reviewer fB8c.

---

### Decision · Program_Chairs · 2026-01-26

Accept (Poster)